# Understanding the Theoretical Generalization Performance of Federated Learning

## Abstract

Federated Learning (FL) has become widely popular because of its applicability in training ML on different sites without data sharing. However, the generalization performance of FL has remained relatively under-explored, primarily due to the intricate interplay between data heterogeneity and the local update procedures intrinsic to FL. This motivates us to answer a fundamental question in FL: *How can we precisely quantify the impact of data heterogeneity and the local update process on the generalization performance for FL as the learning process evolves?* To this end, we conduct a comprehensive theoretical study of FL's generalization performance using a linear model as the first step, where the data heterogeneity is considered for both the stationary and online/non-stationary cases. By providing closed-form expressions of the model error, we rigorously quantify the impact of local update steps (denoted as $K$) under three distinct settings ($K = 1$, $K < \infty$, and $K = \infty$) and how the generalization performance evolves with the round number $t$. Our investigation also provides a comprehensive understanding of how different configurations (including the number of model parameters $p$ and the number of training samples $n$) contribute to the overall generalization performance, thus shedding new insights (such as benign overfitting) for the practical implementation of FL.

## 1 Introduction

Federated Learning (FL) has recently emerged as a prominent paradigm in the realm of distributed learning, facilitating the collaborative training of machine learning models among clients under the orchestration of a central server. By combining privacy preservation, scalability, and collaborative intelligence, FL offers a promising approach for a private, distributed, and efficient machine learning paradigm, revolutionizing industries in healthcare, finance, IoT and various others (Yang et al., 2019b; Xu et al., 2021; Long et al., 2020; Khan et al., 2021). Sparked by the FedAvg algorithm (McMahan et al., 2017), numerous algorithms in FL have demonstrated the ability to achieve fast convergence rates in optimization, thus highlighting the remarkable efficacy of this powerful learning framework. However, the overarching generalization performance of FL remains poorly understood, posing a hurdle to the widespread adoption and practical implementation of FL. The main challenge of understanding FL's generalization performance stems from the distinctive attributes intrinsic to FL: the intricate interplay of *data heterogeneity* and *local update steps*. The data heterogeneity can cause degraded performance with poor generalization in many numerical experiments (Caldarola et al., 2022; Zhao et al., 2018) while other works argue that simple FedAvg algorithm can work very well with data heterogeneity (Wang et al., 2022). On the other hand, existing works have empirically shown that FL algorithms using finetuned local update steps exhibit a better generalization performance than parallel stochastic gradient descent (SGD) algorithm (Lin et al., 2019; Wang & Joshi, 2021; Ortiz et al., 2021). Nevertheless, how to choose the appropriate local update steps for different tasks remains unclear in the literature so far. Given the ever-increasing importance of FL, a compelling open question arises: *How does the data heterogeneity and local update process impact the generalization in FL over the course of learning?*

From a theoretical perspective, there has been relatively limited studies in addressing this question. We can categorize existing explorations into two distinct classes. The first line of work employs the traditional analytical tools from statistical learning, such as the "probably approximately correct" (PAC) framework. These works focus on the domain changes due to the data and system hetero-

geneity. For example, Yuan et al. (2022) and Hu et al. (2023) assume that clients' data distributions are drawn from a meta-population distribution. Accordingly, they define two generalization gaps in FL: one is the participation generalization gap, which measures the difference between the empirical and expected risk for participating clients; the other is the non-participation generalization gap, which measures the difference of the expected risk between participating and non-participating clients. The second class of works studied the training dynamic near a manifold of minima and the effect of stochastic gradient noise on generalization. Caldarola et al. (2022) investigated the generalization behavior through the lens of the geometry of the loss and Hessian eigenspectrum. Gu et al. (2022) utilized the stochastic differential equation (SDE) approximation to study the long-term behavior of the learning process. More recently, Sun et al. (2023) studied FL generalization by data heterogeneity through algorithmic stability and Sefidgaran et al. (2023) established rate-distortion theoretic bounds on FL the generalization. Despite the valuable insights these works offer regarding the generalization performance in FL, it is important to note that they primarily yield asymptotic results by focusing on domain changes or describing asymptotic behavior such as sufficiently large communication rounds and fine-tuned local steps. As a result, these works do not provide an explicit relationship to show how the critical factors in FL, (e.g., the local update process, the number of communication rounds, and data heterogeneity) affect the generalization of FL in general. The intricate interplay between heterogeneous data and local update steps as the learning process evolves (i.e., more communication round) poses a challenge in explicitly characterizing the individual impact of these factors on generalization for FL.

To bridge this gap, as a starting point, we conduct a comprehensive theoretical study of FL's generalization performance using an over-parameterized linear model. Our objective is *to explicitly quantify the influence of data heterogeneity, local update steps, and the total number of communication rounds on the generalization performance of FL.* We highlight our contributions as follows:

- First, we study a FL linear regression model with Gaussian features in both over-parameterized (relates to the study of benign overfitting (Li et al., 2023; Ju et al., 2020; Belkin et al., 2020)) and under-parameterized regimes. At round $t$ of FL, agent $i$ aims to learn a model $\boldsymbol{w}$ through its own local data given by $\boldsymbol{y}_{(i),t} = \mathbf{X}_{(i),t}^{\top} \boldsymbol{w}_{(i),t} + \boldsymbol{\epsilon}_{(i),t}, i \in [m]$. Here $\boldsymbol{w}_{(i),t}$ is the underlying ground-truth parameters that generate the local data. By considering different $\boldsymbol{w}_{(i),t}$, the data samples $(\mathbf{X}_i, \mathbf{y}_i)$ can simulate various patterns of data heterogeneity as the foundation of our study, including both stationary (i.e., $\boldsymbol{w}_{(i),t} = \boldsymbol{w}_{(i)}$) and online/non-stationary (i.e., $\boldsymbol{w}_{(i),t}$ changes over time) cases. By leveraging this model, we can effectively decouple individual effects of the heterogeneous data, the local update process, and the communication round in FL.

- Building upon this model, we provide *closed-form* expressions of the generalization error. Specifically, we rigorously quantify the impact of local update steps (denoted as $K$) under three distinct settings ($K = 1$, $K < \infty$, and $K = \infty$) and show how the generalization performance evolves with the number of communication round $t$. Our results show interesting insights that 1) a good pre-trained model helps but only to some extent; 2) the effect of noise and heterogeneity accumulates but is still limited; 3) the optimal number of local updates sometimes exists; and 4) benign overfitting can exist in FL with alleviated null risk.

## 2 SYSTEM MODEL

### 2.1 LINEAR GROUND TRUTH, PARAMETERS, AND TRAINING SAMPLES

Before considering FL where there are multiple agents, we first introduce the general linear ground truth model which is widely used in the literature of machine learning theory:

$$y = \tilde{\boldsymbol{x}}^{\top} \tilde{\boldsymbol{w}} + \epsilon, \tag{1}$$

where $\tilde{\boldsymbol{x}} \in \mathbb{R}^s$ denotes the feature vector that consists of $s$ true features, $\tilde{\boldsymbol{w}} \in \mathbb{R}^s$ denotes the corresponding true parameters, and $\epsilon \in \mathbb{R}$ denote the noise in the output $y \in \mathbb{R}$. Let $p$ denote the number of features/parameters for the chosen learning model. In other words, a sample is in the form of $(\boldsymbol{x} \in \mathbb{R}^p, y)$. In practice, people usually use a large number of features (may or may not be necessary) to make sure that all true features are included. Thus, we assume that $p \geq s$ and those $p$

features include all necessary features[1]. Without loss of generality, we let $\tilde{\boldsymbol{x}}$ be the first $s$ elements of $\boldsymbol{x}$. Correspondingly, we define $\boldsymbol{w} := \left[\begin{smallmatrix} \tilde{w} \\ \mathbf{0} \end{smallmatrix}\right] \in \mathbb{R}^p$. Thus, Eq. (1) can be rewritten as $y = \boldsymbol{x}^\top \boldsymbol{w} + \epsilon$.

Consider the FL setting where there are $m$ agents and communication rounds indexed by $t = 1, 2, \cdots, T$. We use $[m]$ to denote the set $\{1, 2, \cdots, m\}$, and use $[T]$ to denote the set $1, 2, \cdots, T$. We use the subscript $(\cdot)_{(i),t}$ to denote a quantity for the $i$-th agent at the $t$-th round. In the $t$-th round of FL, the $i$-th agent has $n_{(i),t}$ training samples. Stacking these training samples, we have the following matrix equation.

$$\boldsymbol{y}_{(i),t} = \mathbf{X}_{(i),t}^\top \boldsymbol{w}_{(i),t} + \boldsymbol{\epsilon}_{(i),t}, \tag{2}$$

where $\mathbf{X}_{(i),t} \in \mathbb{R}^{p \times n_{(i),t}}$, $\boldsymbol{w}_{(i),t} \in \mathbb{R}^p$, $\boldsymbol{y}_{(i),t} \in \mathbb{R}^{n_{(i),t}}$, and $\boldsymbol{\epsilon}_{(i),t} \in \mathbb{R}^{n_{(i),t}}$. Usually, in FL, the ground truth parameters of each agent should be close to a common one defined as $\boldsymbol{w}^* \in \mathbb{R}^p$ (and thus $\boldsymbol{w}^*$ should be the target/ideal solution of FL). In other words, in the ideal situation of FL, $\boldsymbol{w}_{(i),t} = \boldsymbol{w}^*$ and does not change with time/round/agents. However, we still keep the subscript $(\cdot)_{(i),t}$ in $\boldsymbol{w}_{(i),t}$ since it is a more general setup and can handle the non-ideal cases such as heterogeneity and non-stationarity.

## 2.2 DATA DISTRIBUTION, HETEROGENEITY, AND NON-STATIONARITY

In order to analytically show the generalization performance of FL, we need some assumptions on the distribution of the training data $\left(\mathbf{X}_{(i),t}, \boldsymbol{y}_{(i),t}\right)_{i \in [m], t=1,2,\cdots,T}$. For tractable theoretical derivation, we adopt independent Gaussian features and noise. Specifically, we have the following assumption.

**Assumption 1.** *For any $i, t$, each element of $\mathbf{X}_{(i),t}$ follows* i.i.d. *standard Gaussian distribution, and each element of $\boldsymbol{\epsilon}_{(i),t}$ follows independent Gaussian distribution with zero mean and variance $\sigma_{(i),t}^2$.*

Since we consider a linear setting, the heterogeneity of the variance of $\mathbf{X}_{(i),t}$ can be normalized, i.e., it is equivalent to only considering the heterogeneity of the variance of $\boldsymbol{\epsilon}_{(i),t}$ as described in Assumption 1. Note that although $\mathbf{X}_{(i),t}$ has identical distribution among different agents, the training data are heterogeneous in $\boldsymbol{y}_{(i),t}$ because $\boldsymbol{w}_{(i),t}$ can be different and $\sigma_{(i),t}$ may have different values. In other words, $\boldsymbol{y}_{(i),t}$ and $\boldsymbol{y}_{(j),t}$ may have different distributions for different $i$ and $j$ in our model.

To quantify the level of heterogeneity in the ground-truth $\boldsymbol{w}_{(i),t}$, we define

$$\boldsymbol{\gamma}_{(i),t} := \boldsymbol{w}^* - \boldsymbol{w}_{(i),t}. \tag{3}$$

Intuitively, $\boldsymbol{\gamma}_{(i),t}$ describes the (small) perturbation of agent $i$'s ground truth at the $t$-th round with respect to the target ground truth $\boldsymbol{w}^*$.

## 2.3 FEDERATED LEARNING PROCESS

We use mean-squared-error (MSE) as the training loss, i.e., the training loss of the parameters $\hat{\boldsymbol{w}}$ on $n$ samples $(\mathbf{X}, \boldsymbol{y})$ is

$$L(\hat{\boldsymbol{w}}; \mathbf{X}, \boldsymbol{y}) := \frac{1}{2n} \left\| \boldsymbol{y} - \mathbf{X}^\top \hat{\boldsymbol{w}} \right\|^2. \tag{4}$$

We consider the FedAvg (McMahan et al., 2017) algorithm, where a central server averages the local updates of each agent (weighted by each agent's number of samples) and then distributes the weighted averaged result to all agents as the initial point of the next local update. We use $\hat{\boldsymbol{w}}_{\text{avg},t} \in \mathbb{R}^p$ to denote the weighted average result at round $t$, and use $\hat{\boldsymbol{w}}_{(i),t} \in \mathbb{R}^p$ to denote the result of the local update of agent $i$ at round $t$. The weighted average can be expressed as:

$$\hat{\boldsymbol{w}}_{\text{avg},t} := \frac{\sum_{i \in [m]} n_{(i),t} \hat{\boldsymbol{w}}_{(i),t}}{\sum_{i \in [m]} n_{(i),t}}. \tag{5}$$

Let $\hat{\boldsymbol{w}}_0$ denote the initialization of the parameters (e.g., by a pre-trained model). For the convenience of notation, we define $\hat{\boldsymbol{w}}_{\text{avg},0} := \hat{\boldsymbol{w}}_0$.

---

[1]Our result can be generalized to the case of missing features by treating the missing part as noise.

One of the focuses of this paper is to examine the impact of local updates. To that end, we use a parameter $K > 0$ to denote the number of local steps, and we consider the following three situations corresponding to different $K$ values: $K = 1$, $K < \infty$, and $K = \infty$. We use superscripts $(\cdot)^{K=1}$, $(\cdot)^{K<\infty}$, and $(\cdot)^{K=\infty}$ to differentiate the notations corresponding to these cases. For example, $\hat{\boldsymbol{w}}_{\text{avg},t}^{K=1}$ and $\hat{\boldsymbol{w}}_{(i),t}^{K=1}$ denotes the value of $\hat{\boldsymbol{w}}_{\text{avg},t}$ and $\hat{\boldsymbol{w}}_{(i),t}$ respectively when we adopt the configuration of $K = 1$.

### 2.3.1 $K = 1$ (ONE-STEP GRADIENT)

The simplest algorithm in FL is to perform only one gradient step in each agent's local update. Specifically, for all agents $i \in [m]$ and each round $t = 1, 2, \cdots, T$, the result of the local step (denoted by $\hat{\boldsymbol{w}}_{(i),t}^{K=1}$) can be written as:

$$\hat{\boldsymbol{w}}_{(i),t}^{K=1} := \hat{\boldsymbol{w}}_{\text{avg},t-1}^{K=1} - \alpha_{(i),t} \frac{\partial L(\hat{\boldsymbol{w}}_{\text{avg},t-1}^{K=1}; \mathbf{X}_{(i),t}, \boldsymbol{y}_{(i),t})}{\partial \hat{\boldsymbol{w}}_{\text{avg},t-1}^{K=1}},$$

where $\alpha_{(i),t} > 0$ denotes agent $i$'s step size (learning rate) of the local update at round $t$.

### 2.3.2 GENERAL $K < \infty$ (MULTI-BATCH LOCAL STEPS)

A more general case is that in each round $t$, every agent can update multiple (finite) times. In the $k$-th update, agent $i$ uses $\tilde{n}_{(i),t}$ data $(\mathbf{X}_{(i),t,k}, \boldsymbol{y}_{(i),t,k})$ (as a batch) where $\mathbf{X}_{(i),t,k} \in \mathbb{R}^{p \times \tilde{n}_{(i),t}}$ and $\boldsymbol{y}_{(i),t,k} \in \mathbb{R}^{\tilde{n}_{(i),t}}$. In this paper, we consider the situation where $\mathbf{X}_{(i),t,k}$ for all $k \in [K]$ are disjoint with each other and their union is $\mathbf{X}_{(i),t}$. In other words, the data $\mathbf{X}_{(i),t}$ are split evenly into $K$ batches (and thus we have $K \cdot \tilde{n}_{(i),t} = n_{(i),t}$). We define $\hat{\boldsymbol{w}}_{(i),t,k}$ as the result after $k$-th batch for the agent $i$ at round $t$. Specifically, for the local update for the $k$-th batch, we have

$$\hat{\boldsymbol{w}}_{(i),t,k} := \hat{\boldsymbol{w}}_{(i),t,k-1} - \alpha_{(i),t} \frac{\partial L(\hat{\boldsymbol{w}}_{(i),t,k-1}; \mathbf{X}_{(i),t,k}, \boldsymbol{y}_{(i),t,k})}{\partial \hat{\boldsymbol{w}}_{(i),t,k-1}}, \quad k = 1, 2, \cdots, K,$$

where $\alpha_{(i),t} > 0$ denotes the learning rate. Notice that $\hat{\boldsymbol{w}}_{(i),t,0} := \hat{\boldsymbol{w}}_{\text{avg},t-1}^{K=1}$ and $\hat{\boldsymbol{w}}_{(i),t} := \hat{\boldsymbol{w}}_{(i),t,K}$. We note that this general case degenerates to that of Section 2.3.1 when $K = 1$.

### 2.3.3 $K = \infty$ (CONVERGENCE IN LOCAL UPDATE)

In this case with $K = \infty$, we consider each agent's solution that the local GD/SGD converges to[2], which is different from Sections 2.3.1 and 2.3.2 where every sample is only trained once. In the under-parameterized regime $p < n_{(i),t}$, the convergence point at each client corresponds to the solution that minimizes the local training loss, i.e.,

$$\hat{\boldsymbol{w}}_{(i),t}^{K=\infty} := \arg\min_{\hat{\boldsymbol{w}}} L(\hat{\boldsymbol{w}}; \mathbf{X}_{(i),t}, \boldsymbol{y}_{(i),t}), \quad \text{when } p < n_{(i),t}.$$

In the over-parameterized regime $p > n_{(i),t}$, there are infinitely many solutions that make the training loss zero with probability 1, i.e., overfitted solutions. It is known in the literature that an overfitted solution corresponding to GD/SGD on a linear model in the over-parameterized regime has the smallest $\ell_2$-norm of the change of parameters (Gunasekar et al., 2018; Lin et al., 2023). Specifically, the convergence point of the local updates corresponds to the solution to the following optimization problem: for $t = 1, 2, \cdots, T$, when $p > n_{(i),t}$, we have

$$\hat{\boldsymbol{w}}_{(i),t}^{K=\infty} := \arg\min_{\hat{\boldsymbol{w}}} \ \left\| \hat{\boldsymbol{w}} - \hat{\boldsymbol{w}}_{\text{avg},t-1}^{K=\infty} \right\|, \quad \text{subject to } \mathbf{X}_{(i),t}^{\top} \hat{\boldsymbol{w}} = \boldsymbol{y}_{(i),t}. \tag{6}$$

The constraint in Eq. (6) implies that the training loss is exactly zero (i.e., overfitted).

## 2.4 GENERALIZATION PERFORMANCE METRIC

We then use the distance between the trained model $\hat{\boldsymbol{w}}$ and the ground truth model $\boldsymbol{w}^*$, i.e., model error, to characterize the generalization performance[3]: $L^{\text{model}}(\hat{\boldsymbol{w}}) = \|\hat{\boldsymbol{w}} - \boldsymbol{w}^*\|^2$. For convenience,

---

[2]The difference between a very large but finite $K$ and the infinite $K$ has been characterized in the literature of the convergence analysis on gradient descent, e.g., Gower (2018); Garrigos & Gower (2023).

[3]We can show that the model error is equal to the expected test error for noise-free data. See Lemma 6.

we define

$$\boldsymbol{\Delta}_t := \boldsymbol{w}^* - \hat{\boldsymbol{w}}_{\text{avg},t}, \qquad t = 0, 1, 2, \cdots, T. \tag{7}$$

Therefore, to characterize the generalization performance of FL at the end of round $t$, we need to quantify $\|\boldsymbol{\Delta}_t\|^2$ with respect to $p$, $K$, $n$, learning rates, initialization, etc. Note that $\boldsymbol{\Delta}_0$ characterizes the difference between the initial weights $\hat{\boldsymbol{w}}_0$ (which can be viewed as a pre-trained model) and the ideal solution $\boldsymbol{w}^*$ (thus $\boldsymbol{\Delta}_0$ is irrelevant to the configuration of $K$).

## 2.5 EXTRA NOTATIONS

Let $\text{seq}_i(\cdot)$ denote a sequence of numbers/vectors (iterating over index $i$). For $l = 1, 2, \cdots$ and considering a real number/vector $\boldsymbol{\beta}_0$, we define a mapping $\mathcal{F}$ as

$$\mathcal{F}(l, \boldsymbol{\beta}_0, \text{seq}_i(a_i), \text{seq}_i(b_i)) := \prod_{i=1}^{l} a_i \boldsymbol{\beta}_0 + \sum_{i=1}^{l} b_i \cdot \prod_{j=i+1}^{l} a_j. \tag{8}$$

Eq. (8) corresponds to the general-term formula of $\boldsymbol{\beta}_l$ for the recurrence relation $\boldsymbol{\beta}_i = a_i \boldsymbol{\beta}_{i-1} + b_i$.

## 3 MAIN RESULTS

In this section, we will present the closed-form expression of $\mathbb{E}\|\boldsymbol{\Delta}_t\|^2$ for all three cases of $K$. These expressions are relatively lengthy since our system model considers both the non-stationarity along different rounds and heterogeneity among different agents. To make our results easy to interpret, we also provide a simplified version by considering a special case, where the non-stationarity and the heterogeneity are constrained. Specifically, the simple case is defined as: for all $i \in [m], t \in [T]$,

$$n_{(i),t} \equiv n, \ \alpha_{(i),t} \equiv \alpha, \ \sigma_{(i),t} \equiv \sigma, \ \sum_{j \in [m]} \boldsymbol{\gamma}_{(j),t} \equiv 0, \ \frac{\sum_{j \in [m]} \|\boldsymbol{\gamma}_{(j),t}\|^2}{m} \equiv \overline{\|\boldsymbol{\gamma}\|^2}, \tag{9}$$

where the symbol $\equiv$ means "always equal to the same constant", and $\overline{\|\boldsymbol{\gamma}\|^2} \geq 0$ denotes the level of heterogeneity. Here $\sum_{j \in [m]} \boldsymbol{\gamma}_{(j),t} \equiv 0$ indicates that the ideal solution $\boldsymbol{w}^*$ is the average of the all agents' ground truth $\boldsymbol{w}_{(i),t}$. We are now ready to present our main results in the following subsections.

### 3.1 $K = 1$

We define the following short-hand notations:

$$\boldsymbol{g}_l^{K=1} := \mathcal{F}(l, \boldsymbol{\Delta}_0, \text{seq}_t\left(\frac{\sum_{i \in [m]} n_{(i),t}(1 - \alpha_{(i),t})}{\sum_{i \in [m]} n_{(i),t}}\right), \text{seq}_t\left(\frac{\sum_{i \in [m]} \alpha_{(i),t} n_{(i),t} \boldsymbol{\gamma}_{(i),t}}{\sum_{i \in [m]} n_{(i),t}}\right)), \tag{10}$$

$$H_t = \frac{\left(\sum_{i \in [m]} n_{(i),t}(1 - \alpha_{(i),t})\right)^2}{\left(\sum_{i \in [m]} n_{(i),t}\right)^2} + \frac{\sum_{i \in [m]} \alpha_{(i),t}^2 n_{(i),t}(p+1)}{\left(\sum_{i \in [m]} n_{(i),t}\right)^2}, \tag{11}$$

$$G_t = \frac{\sum_{i \in [m]} \alpha_{(i),t}^2 p n_{(i),t} \sigma_{(i),t}^2}{\left(\sum_{i \in [m]} n_{(i),t}\right)^2} + \frac{\left\|\sum_{i \in [m]} \alpha_{(i),t} n_{(i),t} \boldsymbol{\gamma}_{(i),t}\right\|^2}{\left(\sum_{i \in [m]} n_{(i),t}\right)^2} + \frac{\sum_{i \in [m]} \alpha_{(i),t}^2 n_{(i),t}(p+1) \|\boldsymbol{\gamma}_{(i),t}\|^2}{\left(\sum_{i \in [m]} n_{(i),t}\right)^2}$$
$$+ \frac{2\left(\sum_{i \in [m]} n_{(i),t}(1 - \alpha_{(i),t})\right) \cdot \left(\sum_{i \in [m]} n_{(i),t} \alpha_{(i),t} \boldsymbol{\gamma}_{(i),t}^\top \boldsymbol{g}_{t-1}\right)}{\left(\sum_{i \in [m]} n_{(i),t}\right)^2}$$
$$- \frac{2 \sum_{i \in [m]} \alpha_{(i),t}^2 n_{(i),t}(p+1) \boldsymbol{\gamma}_{(i),t}^\top \boldsymbol{g}_{t-1}^{K=1}}{\left(\sum_{i \in [m]} n_{(i),t}\right)^2}. \tag{12}$$

**Theorem 1.** *When $K = 1$, we have*

$$\mathbb{E}\left\|\boldsymbol{\Delta}_t^{K=1}\right\|^2 = \mathcal{F}(t, \|\boldsymbol{\Delta}_0\|^2, seq_l(H_l), seq_l(G_l)), \qquad \text{for all } t \in [T]. \tag{13}$$

*For the simple case described by Eq. (9), we have*

$$\mathbb{E}\left\|\boldsymbol{\Delta}_t^{K=1}\right\|^2 = H^t \left\|\boldsymbol{\Delta}_0\right\|^2 + \frac{1-H^t}{1-H}G, \tag{14}$$

*where* $H := (1-\alpha)^2 + \frac{\alpha^2(p+1)}{mn}$, $G := \frac{p\alpha^2\sigma^2}{mn} + \frac{\alpha^2(p+1)}{mn} \cdot \overline{\|\boldsymbol{\gamma}\|^2}$.

We relegate the proof of Theorem 1 to Appendix B. In what follows, we offer two important insights derived from Theorem 1 to discuss the effect of model initialization, data heterogeneity and noise.

**Insight 1) Effect of model initialization: A good pre-trained model helps, but its effect attenuates with the number of communication rounds and it cannot resolve the data heterogeneity challenge.** In Theorem 1, $\|\boldsymbol{\Delta}_0\|^2$ denotes the model error induced by the model initialization $\hat{w}_0$ (cf. Eq. (7)). Our Theorem 1 shows that starting from a good initialization (e.g., a pre-trained model) reduces the training time required to reach a target error rate. The reason is that a good pre-trained model is relatively closer to the target solution $w^*$ than a random model initialization. Thus, $\|\boldsymbol{\Delta}_0\|$ will be small and it helps to reduce the model error. This result theoretically explains previously experimental results that using pre-trained models as the initialization for FL accelerates the training process (Chen et al., 2022; Nguyen et al., 2023). Meanwhile, we note that the coefficient of $\|\boldsymbol{\Delta}_0\|^2$ decreases as $t$ increases when the learning rate is relatively small.[4] It means the effect of the pre-trained model attenuates with the number of communication rounds. As $t \to \infty$, the first term in Eq. (14) asymptotically goes to 0, signifying a diminishing effect of the pre-trained model. This finding is consistent with existing analyses in FL, suggesting that pre-training becomes unnecessary with sufficiently extended training periods (Gu et al., 2022). In addition, Theorem 1 shows the error induced by data noise and heterogeneity remains unaffected by the model initialization. This means even a good pre-trained model can not alleviate the problems caused by heterogeneous data, which aligns with experimental observations (Chen et al., 2022).

**Insight 2) Effect of noise and heterogeneity: Errors arising from data noise and heterogeneity accumulate as the number of communication rounds increases, but eventually converge to an asymptotic limit.** In Eq. (14), the coefficient of the second error term attributed to data noise and heterogeneity ($G$) is expressed as $\frac{1-H^t}{1-H} = 1 + H + H^2 + \cdots + H^{t-1}$. This implies that the error induced by data noise and heterogeneity accumulates with $t$. Meanwhile, this error term does not exhibit unbounded growth; instead, it eventually converges to $\frac{1}{1-H}G$ as $t \to \infty$. This aligns with the prevailing consensus that FL algorithms can perform effectively, despite the occurrence of model drift resulting from data heterogeneity (Wang et al., 2022; Li et al., 2020b;a; Yang et al., 2020).

In Figure 1, we plot the model error with respect to (w.r.t.) $t$ for three different pre-trained models: $\|\boldsymbol{\Delta}_0\| = 1$ (red solid line with markers "□"), $\|\boldsymbol{\Delta}_0\| = 0.5$ (green dashed line with markers "▷"), and $\|\boldsymbol{\Delta}_0\| = 0$ (blue dotted line with markers "○"). The blue curve corresponds to the smallest initial model error and is lower than the other two curves, but the gap diminishes with larger $t$. This phenomenon supports our insights on the effect of model initialization. On the other hand, since the blue curve starts from the ideal solution, its increasing tread w.r.t. $t$ is purely caused by noise and heterogeneity, which also validates our insights on the effect of noise and heterogeneity.

## 3.2 General $K$ (multi-batch local steps)

Similar to Eqs. (11) and (12), we define $\mathcal{J}_l, \mathcal{Q}_l \in \mathbb{R}$. The expressions of $\mathcal{J}_l$ and $\mathcal{Q}_l$ only contain $n_{(i),t}, p, \alpha_{(i),t}, \boldsymbol{\gamma}_{(i),t}, \boldsymbol{\Delta}_0$, and the number of local steps $K$. The formal definitions are provided in Eqs. (50) and (51) at the beginning of Appendix C.

**Theorem 2.** *When $K < \infty$, we have*

$$\mathbb{E}\left\|\boldsymbol{\Delta}_t^{K<\infty}\right\|^2 = \mathcal{F}\left(t, \|\boldsymbol{\Delta}_0\|^2, seq_l(\mathcal{J}_l), seq_l(\mathcal{Q}_l)\right). \tag{15}$$

*For the simple case described by Eq. (9) and by further letting $\overline{\|\boldsymbol{\gamma}\|^2} = 0$, we have*

$$\mathbb{E}\left\|\boldsymbol{\Delta}_t^{K<\infty}\right\|^2 = \mathcal{J}^t\|\boldsymbol{\Delta}_0\|^2 + \frac{1-\mathcal{J}^t}{1-\mathcal{J}} \cdot \frac{\alpha^2 p\sigma^2}{m\tilde{n}} \cdot \frac{1-\mathcal{A}^K}{1-\mathcal{A}}, \tag{16}$$

*where* $\tilde{n} := \lfloor n/K \rfloor, \quad \mathcal{A} := (1-\alpha)^2 + \frac{\alpha^2(p+1)}{\tilde{n}}, \quad \mathcal{J} := \frac{\mathcal{A}^K + (m-1)(1-\alpha)^{2K}}{m}$.

---

[4]In Eq. (14), $H < 1$ when $\alpha_{(i),t} < \frac{2}{1+\frac{p+1}{mn}}$.

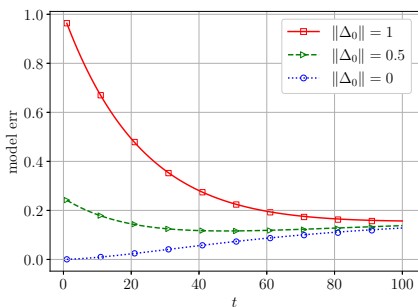

Figure 1: Curves of the model error w.r.t. $t$ where $K = 1$, $m = 3$, $p = 200$, $n_{(i),t} = 50$, $s = 5$, $\left\|\boldsymbol{\gamma}_{(i),t}\right\| = 0.5$, and $\sigma_{(i),t} = 0.7$ for all $i, t$. Each marker point is the average of 20 simulation runs. The curves are theoretical values from Theorem 1. (All markers are close to curves, which validates Theorem 1.)

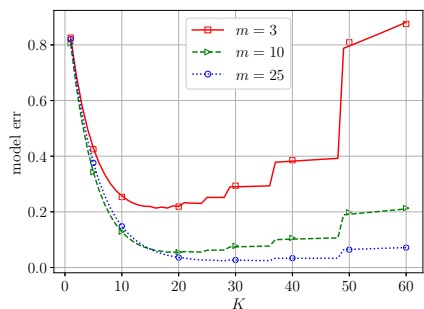

Figure 2: Curves of the model error w.r.t. $K$ where $t = 5$, $s = 5$, $p = 200$, $\|\boldsymbol{\Delta}_0\| = 1$, $n_{(i),t} = 144$, $\left\|\boldsymbol{\gamma}_{(i),t}\right\| = 0.5$, and $\sigma_{(i),t} = 0.7$ for all $i, t$. Each marker point is the average of 20 simulation runs. The curves are theoretical values from Theorem 2. The lowest points of the three curves for cases $m = 3, 10, 25$ are located at $K = 15, 19, 27$, respectively.

The proof for Theorem 2 is provided in Appendix C. Building upon the insights gained from Theorem 2, we have the following discussions concerning the impact of the local update step $K$.

**Insight 3) Effect of the local update step $K$: The optimal choice of finite $K$ sometimes exists.** In Eq. (16), the local update step $K$ simultaneously influences these two error terms, with several factors demonstrating a high correlation with $K$. Therefore, the optimal choice of $K$ should be contingent upon other configurations, such as the number of communication round $t$, $\|\boldsymbol{\Delta}_0\|^2$ (determined by the model initialization), and the noise denoted by $\sigma^2$. Through an analysis of how Eq. (16) evolves with $K$, we establish the following proposition for the optimal choice of $K$:

**Proposition 1.** *Optimal choice of $K$ (defined by $K_{opt}$) for Eq. (16) in different cases are as follows:*

*(1) Finite $K_{opt}$ must exist when $\tilde{n}$ is fixed (i.e., $n$ is determined by $K\tilde{n}$), $\alpha$ is sufficiently small[5], and $t \to \infty$.*

*(2) Finite $K_{opt}$ does not exist (i.e., $K_{opt} = \infty$) when $\tilde{n}$ is fixed, $\alpha$ is sufficiently small, and $\sigma = 0$.*

*(3) When $n$ is fixed (i.e., $\tilde{n}$ is determined by $\lfloor n/K \rfloor$), $t < \infty$, $\alpha \le 0.1$, $m \ge 3$, and $\sigma = 0$, if we neglect the difference between $\lfloor n/K \rfloor$ and $n/K$, then*

$$\frac{n}{p+1}\left(\frac{2}{\alpha} - 1\right) \le K_{opt} \le \frac{n}{p+1}\frac{(m-2)}{\alpha^3}. \tag{17}$$

In Proposition 1, we show that the optimal choice of finite $K$ only exists in some cases, whose value depends on other parameters in one specific problem configuration. For example, the upper bound of $K_{\text{opt}}$ in Eq. (17) indicates that **the optimal $K$ may increase when the number of agents $m$ increases**. This discovery offers insight to interpret experimental observations, wherein switching to local update steps yields divergent outcomes for various tasks; some exhibit improved performance, while others do not (Lin et al., 2019; Ortiz et al., 2021; Gu et al., 2022). Proof of Proposition 1 is provided in Appendix D.

In Figure 2, we plot the model error against $K$ when $n_{(i),t}$ is fixed. The three curves in Figure 2 correspond to different values of $m$. We can see that each of the three curves in Figure 2 has a minimum. The lowest points of the three curves for cases $m = 3, 10, 25$ are located at $K = 15, 19, 27$ (i.e., $K_{\text{opt}}$), respectively. This phenomenon supports our insights that the optimal $K$ can sometimes exist and may increase w.r.t. $m$.

---

[5]When $\alpha < \frac{2}{1 + \frac{p}{\tilde{n}}}$, we have $\mathcal{A} < 1$, and thus $\mathcal{J} < \frac{1 + (m-1)}{m} = 1$.

### 3.3 $K = \infty$ (CONVERGENCE IN LOCAL UPDATE)

We define the following short-hand notations:

$$\boldsymbol{g}_l^{K=\infty} := \mathcal{F}\left(l, \boldsymbol{\Delta}_0, \text{seq}_t\left(\frac{\sum_{i\in[m]} n_{(i),t}\left(1 - \frac{n_{(i),t}}{p}\right)}{\sum_{i\in[m]} n_{(i),t}}\right), \text{seq}_t\left(\frac{\sum_{i\in[m]} \frac{n_{(i),t}^2}{p}\boldsymbol{\gamma}_{(i),t}}{\sum_{i\in[m]} n_{(i),t}}\right)\right), \quad (18)$$

$$C_t := \frac{\sum_{i=1}^m \left(n_{(i),t}^2\left(1 - \frac{n_{(i),t}}{p}\right)\right) + \sum_{i\neq j} n_{(i),t} n_{(j),t}\left(1 - \frac{n_{(i),t}}{p}\right)\left(1 - \frac{n_{(j),t}}{p}\right)}{\left(\sum_{i\in[m]} n_{(i),t}\right)^2}, \quad (19)$$

$$D_t := \frac{\sum_{i\in[m]} \frac{n_{(i),t}^3 \sigma_{(i),t}^2}{p - n_{(i),t} - 1} + \frac{n_{(i),t}^3}{p}\left\|\boldsymbol{\gamma}_{(i),t}\right\|^2}{\left(\sum_{i\in[m]} n_{(i),t}\right)^2}$$

$$+ \frac{\sum_{i\in[m]} \sum_{j\in[m]\setminus\{i\}} \left(\frac{n_{(i),t}^2 n_{(j),t}^2}{p^2}\boldsymbol{\gamma}_{(i),t}^\top\boldsymbol{\gamma}_{(j),t} + 2\frac{n_{(j),t}^2}{p}n_{(i),t}\left(1 - \frac{n_{(i),t}}{p}\right)\boldsymbol{\gamma}_{(j),t}^\top\boldsymbol{g}_{t-1}^{K=\infty}\right)}{\left(\sum_{i\in[m]} n_{(i),t}\right)^2}. \quad (20)$$

**Theorem 3.** *When over-parameterized ($p > \max n_{(i),t} + 1$), we have*

$$\mathbb{E}\left\|\boldsymbol{\Delta}_t^{K=\infty}\right\|^2 = \mathcal{F}(t, \|\boldsymbol{\Delta}_0\|^2, seq_l(C_l), seq_l(D_l)), \qquad \text{for all } t \in [T]. \quad (21)$$

*When under-parameterized ($p < \min n_{(i),t} - 1$), we have*

$$\mathbb{E}\left\|\boldsymbol{\Delta}_t^{K=\infty}\right\|^2 = \left\|\frac{\sum_{i\in[m]} n_{(i),t}\boldsymbol{\gamma}_{(i),t}}{\sum_{i\in[m]} n_{(i),t}}\right\|^2 + \frac{\sum_{i\in[m]} \frac{n_{(i),t}^2 p \sigma_{(i),t}^2}{n_{(i),t}-p-1}}{\left(\sum_{i\in[m]} n_{(i),t}\right)^2}. \quad (22)$$

*For the simple case described by Eq. (9), we have*

$$\mathbb{E}\left\|\boldsymbol{\Delta}_t^{K=\infty}\right\|^2 = \begin{cases} C^t\|\boldsymbol{\Delta}_0\|^2 + \frac{1-C^t}{1-C}D & \text{if over-parameterized,} \\ \frac{p\sigma^2}{m(n-p-1)} & \text{if under-parameterized,} \end{cases} \quad (23)$$

*where*

$$C := \frac{1}{m}\left(1 - \frac{n}{p}\right) + \frac{m-1}{m}\left(1 - \frac{n}{p}\right)^2 < 1, \qquad D := \frac{n\sigma^2}{m(p-n-1)} + \frac{n}{p}\overline{\|\boldsymbol{\gamma}\|^2}. \quad (24)$$

Proof of Theorem 3 is in Appendix E.

**Insight 4) Benign overfitting exists in FL, and the "null risk" is alleviated by using more communications rounds.** In the over-parameterized case of Eq. (23), the term $D$ decreases when $p$ increases. Thus, when the term $D$ dominates (e.g., when noise and/or heterogeneity is large, or $t$ is large), the generalization performance of FL in this case will benefit from more parameters when overfitted. This validates the "double-descent" or benign overfitting phenomenon in the literature of the classical (single-task single-agent) linear regression (e.g. Belkin et al. (2020)). For the comparable Gaussian models we used, the expectation of the model error of such a classical (single-task single-agent) linear regression is

$$(1 - \frac{n}{p})\|\boldsymbol{\Delta}_0\|^2 + \frac{n\sigma^2}{p-n-1}. \quad (25)$$

By Eq. (25) and related literature (e.g., Ju et al. (2020)), the classical linear regression suffers from "null risk" (i.e., converges to the initial error) when $p \to \infty$. However, for the FL result in Eq. (23), we can see that the "null risk" term $\|\boldsymbol{\Delta}_0\|^2$ is alleviated by the coefficient $C^t$ which approaches zero when $t \to \infty$. In other words, for fixed $n$, when $p \to \infty$, as long as we let $t \to \infty$ in a faster speed (e.g., $t = p\log p$, proved in Lemma 1 in Appendix A), then the null risk term $C^t\|\boldsymbol{\Delta}_0\|^2 \to 0$, which implies that using more communication rounds in FL (i.e., larger $t$) mitigates the null risk and thus enhances the benefits of overfitting.

In Figure 3, we plot the model error against $p$ for both the underparameterized regime ($p < n = 25$) and overparameterized regime ($p > 25$) for cases of $t = 1$, $t = 4$, and $t = 40$. We can see that all three curves decrease at the beginning of the overparameterized regime, which validates the existence of benign overfitting. Meanwhile, the curve of $t = 40$ (blue dotted one with markers "$\circ$") has a lower and wider descent curve, which validates our insight that larger $t$ enhances the benefits of overfitting in FL.

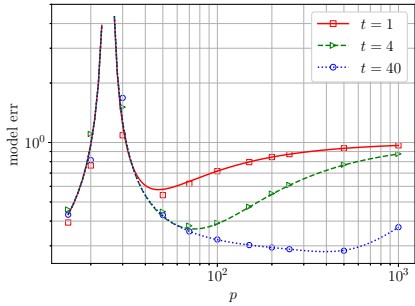

Figure 3: Curves of the model error w.r.t. $p$ where $m = 3$, $s = 5$, $n_{(i),t} = 25$, $\|\mathbf{\Delta}_0\| = 1$, $\|\mathbf{\gamma}_{(i),t}\| = 0.5$, and $\sigma_{(i),t} = 0.7$ for all $i, t$. Each marker is the average of 20 simulation runs. The curves are theoretical values from Theorem 3.

# 4 RELATED WORK

In the literature, there has been relatively limited studies on the generalization of FL. We categorize these works into three distinct classes. The first line of works employs the traditional analytical tools from statistical learning. Yuan et al. (2022) assumes that clients' data distributions are drawn from a meta-population distribution. Accordingly, they define two generalization gaps in FL: one is the participation generalization gap to measure the difference between the empirical and expected risk for participating clients, the same as the definition in classic statistical learning; the second is the non-participation generalization gap, which measures the difference of the expected risk between participating and non-participating clients. Following this two-level distribution framework, sharper bounds are provided (Hu et al., 2023). Zhao et al. (2023) utilized the Probably Approximately Correct (PAC) Bayesian framework to investigate a tailored generalization bound for heterogeneous data in FL. More works utilize similar tools to study the generalization in FL (Chor et al., 2023; Barnes et al., 2022; Sefidgaran et al., 2022; Sun et al., 2023; Sefidgaran et al., 2023; Huang et al., 2021). The second class of works studied the training dynamic near a manifold of minima and the effect of stochastic gradient noise on generalization. They used "sharpness" as a useful tool for generalization. Caldarola et al. (2022) and Shi et al. (2023) investigated the generalization behavior through the lens of the geometry of the loss and Hessian eigenspectrum, linking the model's lack of generalization capacity to the sharpness of the solution under ideal client participation. Based on the sharpness, Qu et al. (2022) proposed a momentum algorithm with better generalization. Gu et al. (2022) utilizes the stochastic differential equation (SDE) approximation to study the long-term behavior of the learning process. They showed that utilizing local steps always exhibits better generalization under appropriate conditions, including a sufficiently small learning rate, enough number of communication rounds, and the local steps being tuned. All of these existing studies primarily yield asymptotic results by focusing on domain changes or describing limiting behavior such as sufficiently large communication rounds and fine-tuned local steps. Consequently, they do not establish a direct, quantifiable relationship that demonstrates how key factors—namely, data heterogeneity, the local update process, and the communication round—affect the generalization performance of FL. In this paper, we achieve the explicit quantification of the impact of data heterogeneity, local update steps, and the total number of communication rounds on the generalization performance within the context of a federated linear regression model.

# 5 CONCLUSION

In this work, we analyze the generalization performance of FL using a linear model (possibly over-parameterized), focusing on the influence of data heterogeneity, local updates, and communication rounds. By providing the closed-form expressions of the model error, we show useful insights that can be used to theoretically explain some interesting phenomena observed in the practice of FL, e.g., a good pre-trained model helps FL's performance to some extent.

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

# Supplemental Material

We give a table to summarize the content of the supplemental material.

| Section | Content |
|---|---|
| Appendix A | some useful lemmas as technical tools |
| Appendix B | proof of Theorem 1 for $K = 1$ |
| Appendix C | proof of Theorem 2 for $K < \infty$ |
| Appendix D | proof of Proposition 1 about optimal $K$ |
| Appendix E | proof of Theorem 3 for $K = \infty$ |
| Appendix F | a table for some important notations |
| Appendix G | more related work |

Table 1: Outline of the supplemental material.

## A  USEFUL LEMMAS

In this section, we provide some useful lemmas. Specifically, Lemma 1 is used to support the claim of the convergence speed in Insight 4. Lemmas 2 to 4 are some results about the Gaussian random matrices that can be found in the literature. We want to highlight Lemma 5 as part of our technical novelty, which gives the exact values of terms related to the projection formed by each agent's training inputs. Lemma 6 is used to justify the definition of model error.

**Lemma 1.** *Recalling the definition of $C$ in Eq.* (24)*, we have*

$$\lim_{t=p\ln p,\ p\to\infty} C^t = 0.$$

*Proof.* We have $C^t \geq 0$ and

$$C^t \leq \left(1 - \frac{n}{p}\right)^t \quad \text{(since } C \leq \left(1 - \frac{n}{p}\right) \text{ because } \left(1 - \frac{n}{p}\right)^2 \leq \left(1 - \frac{n}{p}\right)\text{)}$$

$$= \left(1 + \frac{1}{\frac{p}{n} - 1}\right)^{-t} \quad \text{(since } 1 - \frac{n}{p} = \frac{1}{1 + \frac{1}{\frac{p}{n}-1}}\text{)}$$

$$= \left(1 + \frac{1}{\frac{p}{n} - 1}\right)^{-p\ln p} \quad \text{(since } t = p\ln p\text{)}$$

$$= \left(1 + \frac{1}{\frac{p}{n} - 1}\right)^{-\frac{p}{n}\cdot n\cdot\ln p}$$

$$\leq \left(1 + \frac{1}{\frac{p}{n} - 1}\right)^{-\left(\frac{p}{n}-1\right)\cdot n\cdot\ln p}.$$

Notice that

$$\lim_{p\to\infty} \left(1 + \frac{1}{\frac{p}{n} - 1}\right)^{-\left(\frac{p}{n}-1\right)\cdot n\cdot\ln p} = \lim_{p\to\infty} e^{-n\ln p} = 0,$$

where we use the fact that $\lim_{x\to\infty}(1 + x^{-1})^x = e$. The result of this lemma thus follows by the squeeze theorem. $\qquad\square$

The result of the following lemma can be found in the literature (e.g., (Belkin et al., 2020; Ju et al., 2022)).

**Lemma 2.** *Consider a random matrix $\mathbf{K} \in \mathbb{R}^{p\times n}$ where $p$ and $n$ are two positive integers and $p > n + 1$. Each element of $\mathbf{K}$ is* i.i.d. *according to standard Gaussian distribution. For any fixed*

*vector $\boldsymbol{a} \in \mathbb{R}^p$, we must have*

$$\mathbb{E} \left\| \left( \mathbf{I}_p - \mathbf{K} \left( \mathbf{K}^\top \mathbf{K} \right)^{-1} \mathbf{K}^\top \right) \boldsymbol{a} \right\|^2 = \left( 1 - \frac{n}{p} \right) \|\boldsymbol{a}\|^2 ,$$

$$\mathbb{E} \left\| \mathbf{K} \left( \mathbf{K}^\top \mathbf{K} \right)^{-1} \mathbf{K}^\top \boldsymbol{a} \right\|^2 = \frac{n}{p} \|\boldsymbol{a}\|^2 .$$

The following lemma can be found in Lemma 8 of (Ju et al., 2023).

**Lemma 3.** *Consider a random matrix* $\mathbf{K} \in \mathbb{R}^{a \times b}$ *where* $a > b + 1$. *Each element of* $\mathbf{K}$ *is* i.i.d. *following standard Gaussian distribution* $\mathcal{N}(0,1)$. *Consider three Gaussian random vectors* $\boldsymbol{\alpha}, \boldsymbol{\gamma} \in \mathbf{R}^a$ *and* $\boldsymbol{\beta} \in \mathbf{R}^b$ *such that* $\boldsymbol{\alpha} \sim \mathcal{N}(\mathbf{0}, \sigma_\alpha^2 \mathbf{I}_a)$, $\boldsymbol{\gamma} \sim \mathcal{N}(\mathbf{0}, \mathrm{diag}(d_1^2, d_2^2, \cdots, d_a^2))$, *and* $\boldsymbol{\beta} \sim \mathcal{N}(\mathbf{0}, \sigma_\beta^2 \mathbf{I}_b)$. *Here* $\mathbf{K}$, $\boldsymbol{\alpha}$, $\boldsymbol{\gamma}$, *and* $\boldsymbol{\beta}$ *are independent of each other. We then must have*

$$\mathbb{E} \left[ (\mathbf{K}^\top \mathbf{K})^{-1} \right] = \frac{\mathbf{I}_b}{a - b - 1}, \tag{26}$$

$$\mathbb{E} \left\| \mathbf{K}(\mathbf{K}^\top \mathbf{K})^{-1} \boldsymbol{\beta} \right\|^2 = \frac{b \sigma_\beta^2}{a - b - 1}, \tag{27}$$

$$\mathbb{E} \left\| (\mathbf{K}^\top \mathbf{K})^{-1} \mathbf{K}^\top \boldsymbol{\alpha} \right\|^2 = \frac{b \sigma_\alpha^2}{a - b - 1}, \tag{28}$$

$$\mathbb{E} \left\| (\mathbf{K}^\top \mathbf{K})^{-1} \mathbf{K}^\top \boldsymbol{\gamma} \right\|^2 = \frac{b \sum_{i=1}^a d_i^2}{a(a - b - 1)}. \tag{29}$$

The following lemma can be found in (Bernacchia, 2021) and Lemma 13 of (Ju et al., 2022).

**Lemma 4.** *Consider a random matrix* $\mathbf{K} \in \mathbb{R}^{a \times b}$ *whose each element follows* i.i.d. *standard Gaussian distribution (i.e.,* i.i.d. $\mathcal{N}(0,1)$*). We mush have*

$$\mathbb{E}[\mathbf{K}^\top \mathbf{K}] = a \mathbf{I}_b,$$

$$\mathbb{E}[\mathbf{K} \mathbf{K}^\top] = b \mathbf{I}_a,$$

$$\mathbb{E}[\mathbf{K} \mathbf{K}^\top \mathbf{K} \mathbf{K}^\top] = b(b + a + 1) \mathbf{I}_a.$$

**Lemma 5.** *For any* $i \in [m]$ *and* $t$, *we must have*

$$\mathbb{E}_{\mathbf{P}_{(i),t}} \left[ \mathbf{P}_{(i),t} \boldsymbol{\Delta}_{t-1}^{K=\infty} \right] = \frac{n_{(i),t}}{p} \boldsymbol{\Delta}_{t-1}^{K=\infty}. \tag{30}$$

*Consequently, when* $i \neq j$, *we have*

$$\mathbb{E}_{\mathbf{P}_{(i),t}, \mathbf{P}_{(j),t}} \left[ \boldsymbol{\Delta}_{t-1}^{K=\infty \top} \mathbf{P}_{(i),t} \mathbf{P}_{(j),t} \boldsymbol{\Delta}_{t-1}^{K=\infty} \right] = \frac{n_{(j),t} n_{(i),t}}{p^2} \left\| \boldsymbol{\Delta}_{t-1}^{K=\infty} \right\|^2 .$$

Before we provide the rigorous proof of Lemma 5, we provide an intuition as follows.

*Intuition of Proof of Lemma 5:* We use Figure 4 to help illustrating the intuition. In Figure 4, the vector $\overrightarrow{OA}$ denotes $\boldsymbol{\Delta}_{t-1}^{K=\infty}$, the plane $\alpha$ denotes the space spanned by the columns of $\mathbf{X}_{(i),t}$. Notice that $\mathbf{P}_{(i),t} \boldsymbol{\Delta}_{t-1}^{K=\infty}$ represents result of projecting $\boldsymbol{\Delta}_{t-1}^{K=\infty}$ to the column space of $\mathbf{X}_{(i),t}$, i.e., the vector $\overrightarrow{OB}$ in Figure 4. Therefore, in Figure 4, calculating $\mathbb{E}_{\mathbf{P}_{(i),t}} \mathbf{P}_{(i),t} \boldsymbol{\Delta}_{t-1}^{K=\infty}$ means calculating the average of $\overrightarrow{OB}$ when the hyper-plane $\alpha$ rotating around the point $O$. Notice that $\overrightarrow{OB} = \overrightarrow{OC} + \overrightarrow{CB}$ where $\overrightarrow{OC}$ and $\overrightarrow{CB}$ are the parallel and perpendicular components of $\overrightarrow{OB}$ w.r.t. $\overrightarrow{OA}$, respectively. Because of the rotational symmetry of the hyper-plane $\alpha$ (due to the rotational symmetry of each column of $\mathbf{X}_{(i),t}$), we know that all the perpendicular components are cancelled out while only the parallel components remain in the averaging process. In other words, for any hyper-plane $\alpha$, there exists a symmetrical (w.r.t. $\overrightarrow{OA}$) hyper-plane $\beta$ with the same probability density such that the projection of $\overrightarrow{OA}$ to the hyper-plane $\beta$, named $\overrightarrow{OB'}$, has the same parallel component $\overrightarrow{OC}$ but the opposite perpendicular component $\overrightarrow{CB'} = -\overrightarrow{CB}$. Thus, we only need to calculate the average of the parallel component $\overrightarrow{OC}$, whose length equals $\cos\theta \left| \overrightarrow{OB} \right|$, where $\theta = \angle AOB$ is defined as

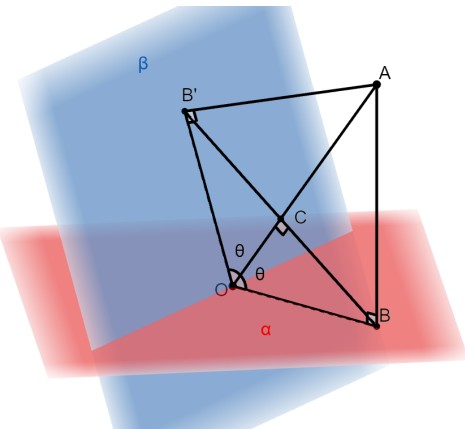

Figure 4: Geometric interpretation of the proof of Lemma 5.

the angle between $\boldsymbol{\Delta}_{t-1}^{K=\infty}$ and $\mathbf{P}_{(i),t}\boldsymbol{\Delta}_{t-1}^{K=\infty}$ (i.e., the angle between $\boldsymbol{\Delta}_{t-1}^{K=\infty}$ and the hyperplane spanned by the columns of $\mathbf{X}_{(i),t}$ as

$$\theta := \arccos \frac{\mathbf{P}_{(i),t}\boldsymbol{\Delta}_{t-1}^{K=\infty}}{\left\|\boldsymbol{\Delta}_{t-1}^{K=\infty}\right\|}. \tag{31}$$

Also notice that $\left|\overrightarrow{OB}\right| = \cos\theta \left|\overrightarrow{OA}\right|$. Thus, the length of the parallel component equals $\left|\overrightarrow{OC}\right| = \cos^2\theta \left|\overrightarrow{OA}\right|$. Therefore, we have $\mathbb{E}\,\overrightarrow{OC} = \mathbb{E}\cos^2\theta\overrightarrow{OA} = \frac{n_{(i),t}}{p}\boldsymbol{\Delta}_{t-1}^{K=\infty}$. The last equation uses Lemma 2.

*Proof.* Let $C := \left\|\boldsymbol{\Delta}_{t-1}^{K=\infty}\right\|$. Since we are calculating expected projection of $\boldsymbol{\Delta}_{t-1}^{K=\infty}$ onto the column space of $\mathbf{X}_{(i),t}$, by the symmetry of $\mathbf{X}_{(i),t}$, without loss of generality we let

$$\boldsymbol{\Delta}_{t-1}^{K=\infty} = C \cdot \begin{bmatrix} 1 \\ 0 \\ 0 \\ \vdots \\ 0 \end{bmatrix}. \tag{32}$$

Define

$$\tilde{\mathbf{X}}_{(i),t} := \begin{bmatrix} -1 & & & \\ & 1 & & \\ & & \ddots & \\ & & & 1 \end{bmatrix} \mathbf{X}_{(i),t}. \tag{33}$$

Since each element of $\mathbf{X}_{(i),t}$ follows *i.i.d.* standard Gaussian distribution, we know that $\tilde{\mathbf{X}}_{(i),t}$ and $\mathbf{X}_{(i),t}$ has identical distributioin. Thus, we have

$$\int \mathbf{X}_{(i),t}(\mathbf{X}_{(i),t}^{\top}\mathbf{X}_{(i),t})\mathbf{X}_{(i),t}^{\top}\boldsymbol{\Delta}_{t-1}^{K=\infty}d\mu(\mathbf{X}_{(i),t}) = \int \tilde{\mathbf{X}}_{(i),t}(\tilde{\mathbf{X}}_{(i),t}^{\top}\tilde{\mathbf{X}}_{(i),t})\tilde{\mathbf{X}}_{(i),t}\boldsymbol{\Delta}_{t-1}^{K=\infty}d\mu(\mathbf{X}_{(i),t}), \tag{34}$$

where $\mu(\mathbf{X}_{(i),t})$ denotes the joint probability distribution of $\mathbf{X}_{(i),t}$.

By Eq. (33), we have

$$\tilde{\mathbf{X}}_{(i),t}^{\top}\tilde{\mathbf{X}}_{(i),t} = \mathbf{X}_{(i),t}^{\top}\begin{bmatrix} -1 & & & \\ & 1 & & \\ & & \ddots & \\ & & & 1 \end{bmatrix}\begin{bmatrix} -1 & & & \\ & 1 & & \\ & & \ddots & \\ & & & 1 \end{bmatrix}\mathbf{X}_{(i),t} = \mathbf{X}_{(i),t}^{\top}\mathbf{X}_{(i),t},$$

$\mathbf{X}_{(i),t}^{\top}\boldsymbol{\Delta}_{t-1}^{K=\infty} = [\mathbf{X}_{(i),t}]_{1,:}, \ \tilde{\mathbf{X}}_{(i),t}^{\top}\boldsymbol{\Delta}_{t-1}^{K=\infty} = -[\mathbf{X}_{(i),t}]_{1,:}$ (here $[\cdot]_{1,:}$ denotes the first row of a matrix).

Thus, we have

$$\tilde{\mathbf{X}}_{(i),t}(\tilde{\mathbf{X}}_{(i),t}^\top \tilde{\mathbf{X}}_{(i),t})^{-1}\tilde{\mathbf{X}}_{(i),t}^\top \boldsymbol{\Delta}_{t-1}^{K=\infty} = -\tilde{\mathbf{X}}_{(i),t}(\tilde{\mathbf{X}}_{(i),t}^\top \tilde{\mathbf{X}}_{(i),t})^{-1}\mathbf{X}_{(i),t}^\top \boldsymbol{\Delta}_{t-1}^{K=\infty}. \tag{35}$$

Therefore, we have

$$\begin{aligned}
&\mathbf{X}_{(i),t}(\mathbf{X}_{(i),t}^\top \mathbf{X}_{(i),t})^{-1}\mathbf{X}_{(i),t}^\top \boldsymbol{\Delta}_{t-1}^{K=\infty} + \tilde{\mathbf{X}}_{(i),t}(\tilde{\mathbf{X}}_{(i),t}^\top \tilde{\mathbf{X}}_{(i),t})^{-1}\tilde{\mathbf{X}}_{(i),t}^\top \boldsymbol{\Delta}_{t-1}^{K=\infty} \\
=& (\mathbf{X}_{(i),t} - \tilde{\mathbf{X}}_{(i),t})(\mathbf{X}_{(i),t}^\top \mathbf{X}_{(i),t})^{-1}\mathbf{X}_{(i),t}^\top \boldsymbol{\Delta}_{t-1}^{K=\infty} \quad \text{(by Eq. (35))} \\
=& \begin{bmatrix} 2 & & & \\ & 0 & & \\ & & \ddots & \\ & & & 0 \end{bmatrix} \mathbf{X}_{(i),t}(\mathbf{X}_{(i),t}^\top \mathbf{X}_{(i),t})^{-1}\mathbf{X}_{(i),t}^\top \boldsymbol{\Delta}_{t-1}^{K=\infty} \quad \text{(by Eq. (33))} \\
=& \begin{bmatrix} 1 \\ 0 \\ \vdots \\ 0 \end{bmatrix} \begin{bmatrix} 2 & 0 & \cdots & 0 \end{bmatrix} \mathbf{X}_{(i),t}(\mathbf{X}_{(i),t}^\top \mathbf{X}_{(i),t})^{-1}\mathbf{X}_{(i),t}^\top \boldsymbol{\Delta}_{t-1}^{K=\infty} \\
=& 2\frac{\boldsymbol{\Delta}_{t-1}^{K=\infty}}{C^2}{\boldsymbol{\Delta}_{t-1}^{K=\infty}}^\top \mathbf{X}_{(i),t}(\mathbf{X}_{(i),t}^\top \mathbf{X}_{(i),t})^{-1}\mathbf{X}_{(i),t}^\top \boldsymbol{\Delta}_{t-1}^{K=\infty} \quad \text{(by Eq. (32))} \\
=& 2\frac{\boldsymbol{\Delta}_{t-1}^{K=\infty}}{C^2}{\boldsymbol{\Delta}_{t-1}^{K=\infty}}^\top \mathbf{P}_{(i),t}\boldsymbol{\Delta}_{t-1}^{K=\infty} \quad \text{(by Eq. (68))} \\
=& 2\frac{\boldsymbol{\Delta}_{t-1}^{K=\infty}}{C^2}\left\|\mathbf{P}_{(i),t}\boldsymbol{\Delta}_{t-1}^{K=\infty}\right\|^2 \quad \text{(since } \mathbf{P}_{(i),t}^\top\mathbf{P}_{(i),t} = \mathbf{P}_{(i),t} \text{ as } \mathbf{P}_{(i),t} \text{ is an orthogonal projection).}
\end{aligned} \tag{36}$$

Thus, we have

$$\begin{aligned}
&\underset{\mathbf{X}_{(i),t}}{\mathbb{E}}\left[\mathbf{P}_{(i),t}\boldsymbol{\Delta}_{t-1}^{K=\infty}\right] \\
=& \int \mathbf{X}_{(i),t}(\mathbf{X}_{(i),t}^\top \mathbf{X}_{(i),t})^{-1}\mathbf{X}_{(i),t}^\top \boldsymbol{\Delta}_{t-1}^{K=\infty} d\mu(\mathbf{X}_{(i),t}) \\
=& \frac{1}{2}\int \left(\mathbf{X}_{(i),t}(\mathbf{X}_{(i),t}^\top \mathbf{X}_{(i),t})^{-1}\mathbf{X}_{(i),t}^\top \boldsymbol{\Delta}_{t-1}^{K=\infty} + \tilde{\mathbf{X}}_{(i),t}(\tilde{\mathbf{X}}_{(i),t}^\top \tilde{\mathbf{X}}_{(i),t})\tilde{\mathbf{X}}_{(i),t}^\top \boldsymbol{\Delta}_{t-1}^{K=\infty}\right) d\mu(\mathbf{X}_{(i),t}) \quad \text{(by Eq. (34))} \\
=& \int \frac{\boldsymbol{\Delta}_{t-1}^{K=\infty}}{C^2}\left\|\mathbf{P}_{(i),t}\boldsymbol{\Delta}_{t-1}^{K=\infty}\right\|^2 d\mu(\mathbf{X}_{(i),t}) \\
=& \frac{\boldsymbol{\Delta}_{t-1}^{K=\infty}}{C^2}\underset{\mathbf{X}_{(i),t}}{\mathbb{E}}\left\|\mathbf{P}_{(i),t}\boldsymbol{\Delta}_{t-1}^{K=\infty}\right\|^2 \\
=& \frac{n_{(i),t}}{p}\boldsymbol{\Delta}_{t-1}^{K=\infty} \quad \text{(by Lemma 2).}
\end{aligned}$$

The result of this lemma thus follows. $\square$

**Lemma 6.** *Let the noise in every test sample have zero mean and variance $\sigma^2$. For any learning result $\hat{\boldsymbol{w}}$, the mean square test error must equal to $\|\hat{\boldsymbol{w}} - \boldsymbol{w}^*\|^2 + \sigma^2$. Therefore, the mean squared test error for noise-free test samples equals to the model error $L^{model}(\hat{\boldsymbol{w}}) = \|\hat{\boldsymbol{w}} - \boldsymbol{w}^*\|^2$.*

*Proof.* Considering $(\boldsymbol{x}, y)$ as a randomly generated test sample by the ground truth $y = \boldsymbol{x}^\top \boldsymbol{w}^* + \epsilon$, the mean squared error is equal to

$$
\begin{aligned}
\mathbb{E}_{\boldsymbol{x},y} \left\| \boldsymbol{x}^\top \hat{\boldsymbol{w}} - y \right\| &= \mathbb{E}_{\boldsymbol{x},\epsilon} \left\| \boldsymbol{x}^\top \hat{\boldsymbol{w}} - (\boldsymbol{x}^\top \boldsymbol{w}^* + \epsilon) \right\|^2 \\
&= \mathbb{E}_{\boldsymbol{x},\epsilon} \left\| \boldsymbol{x}^\top (\hat{\boldsymbol{w}} - \boldsymbol{w}^*) + \epsilon \right\|^2 \\
&= \mathbb{E}_{\boldsymbol{x}} \left\| \boldsymbol{x}^\top (\hat{\boldsymbol{w}} - \boldsymbol{w}^*) \right\|^2 + \mathbb{E}_{\epsilon} \left\| \epsilon \right\|^2
\end{aligned}
$$

(since the noise $\epsilon$ has zero mean and is independent of other random variables)

$$
= \left\| \hat{\boldsymbol{w}} - \boldsymbol{w}^* \right\|^2 + \sigma^2
$$

(notice that $\boldsymbol{x}$ follows standard Gaussian distribution and is independent of $\hat{\boldsymbol{w}}$).

$\square$

## B  PROOF OF THEOREM 1

Calculating the gradient of the training loss defined at Eq. (4), we have

$$
\begin{aligned}
\frac{\partial L(\hat{\boldsymbol{w}})}{\partial \hat{\boldsymbol{w}}} &= \frac{\partial (\boldsymbol{y} - \mathbf{X}^\top \hat{\boldsymbol{w}})}{\partial \hat{\boldsymbol{w}}} \cdot \frac{\partial \frac{1}{2n} \left\| \boldsymbol{y} - \mathbf{X}^\top \hat{\boldsymbol{w}} \right\|^2}{\partial (\boldsymbol{y} - \mathbf{X}^\top \hat{\boldsymbol{w}})} \quad \text{(by the chain rule)} \\
&= -\mathbf{X} \cdot \frac{1}{n} (\boldsymbol{y} - \mathbf{X}^\top \hat{\boldsymbol{w}}) \\
&= \frac{1}{n} (\mathbf{X}\mathbf{X}^\top \hat{\boldsymbol{w}} - \mathbf{X}\boldsymbol{y}).
\end{aligned}
$$

When $K = 1$, with step size $\alpha_{(i),t} > 0$, we thus have

$$
\begin{aligned}
\hat{\boldsymbol{w}}_{(i),t}^{K=1} &= \left( \mathbf{I}_p - \frac{\alpha_{(i),t}}{n_{(i),t}} \mathbf{X}_{(i),t} \mathbf{X}_{(i),t}^\top \right) \hat{\boldsymbol{w}}_{\text{avg},t-1}^{K=1} + \frac{\alpha_{(i),t}}{n_{(i),t}} \mathbf{X}_{(i),t} \boldsymbol{y}_{(i),t} \\
&= \left( \mathbf{I}_p - \frac{\alpha_{(i),t}}{n_{(i),t}} \mathbf{X}_{(i),t} \mathbf{X}_{(i),t}^\top \right) \hat{\boldsymbol{w}}_{\text{avg},t-1}^{K=1} + \frac{\alpha_{(i),t}}{n_{(i),t}} \mathbf{X}_{(i),t} \left( \mathbf{X}_{(i),t}^\top \boldsymbol{w}_{(i),t} + \boldsymbol{\epsilon}_{(i),t} \right) \quad \text{(by Eq. (2))}.
\end{aligned}
$$

Thus, we have

$$
\begin{aligned}
\hat{\boldsymbol{w}}_{\text{avg},t}^{K=1} &= \frac{1}{\sum_{i \in [m]} n_{(i),t}} \sum_{i \in [m]} n_{(i),t} \hat{\boldsymbol{w}}_{(i),t}^{K=1} \\
&= \hat{\boldsymbol{w}}_{\text{avg},t-1}^{K=1} + \frac{1}{\sum_{i \in [m]} n_{(i),t}} \sum_{i \in [m]} \alpha_{(i),t} \left( -\mathbf{X}_{(i),t} \mathbf{X}_{(i),t}^\top \hat{\boldsymbol{w}}_{\text{avg},t-1}^{K=1} + \mathbf{X}_{(i),t} \mathbf{X}_{(i),t}^\top \boldsymbol{w}_{(i),t} + \mathbf{X}_{(i),t} \boldsymbol{\epsilon}_{(i),t} \right).
\end{aligned}
$$

(37)

By Eqs. (3) and (7), we have

$$
\begin{aligned}
&\boldsymbol{\Delta}_t^{K=1} \\
&= \boldsymbol{\Delta}_{t-1}^{K=1} + \frac{1}{\sum_{i \in [m]} n_{(i),t}} \sum_{i \in [m]} \alpha_{(i),t} \left( \mathbf{X}_{(i),t} \mathbf{X}_{(i),t}^\top (\boldsymbol{\gamma}_{(i),t} - \boldsymbol{\Delta}_{t-1}^{K=1}) - \mathbf{X}_{(i),t} \boldsymbol{\epsilon}_{(i),t} \right) \\
&= \frac{1}{\sum_{i \in [m]} n_{(i),t}} \sum_{i \in [m]} \Bigg( \underbrace{\left( n_{(i),t} \mathbf{I}_p - \alpha_{(i),t} \mathbf{X}_{(i),t} \mathbf{X}_{(i),t}^\top \right) \boldsymbol{\Delta}_{t-1}^{K=1}}_{\boldsymbol{q}_{1i}} + \underbrace{\alpha_{(i),t} \mathbf{X}_{(i),t} \mathbf{X}_{(i),t}^\top \boldsymbol{\gamma}_{(i),t}}_{\boldsymbol{q}_{2i}} - \underbrace{\alpha_{(i),t} \mathbf{X}_{(i),t} \boldsymbol{\epsilon}_{(i),t}}_{\boldsymbol{q}_{3i}} \Bigg)
\end{aligned}
$$

(38)

$\left( \text{since } \boldsymbol{\Delta}_{t-1}^{K=1} = \frac{1}{\sum_{i \in [m]} n_{(i),t}} \sum_{i \in [m]} n_{(i),t} \boldsymbol{\Delta}_{t-1}^{K=1} \right).$

Considering the three types of terms $\boldsymbol{q}_{1i}, \boldsymbol{q}_{2i}, \boldsymbol{q}_{3i}$ defined in Eq. (38), by Assumption 1, we have

$$
\begin{aligned}
\underset{t}{\mathbb{E}}\, \boldsymbol{q}_{1i} &= n_{(i),t}\left(1 - \alpha_{(i),t}\right) \boldsymbol{\Delta}_{t-1}^{K=1}, \\
\underset{t}{\mathbb{E}}\, \boldsymbol{q}_{2i} &= \alpha_{(i),t} n_{(i),t} \boldsymbol{\gamma}_{(i),t}, \\
\underset{t}{\mathbb{E}}\, \boldsymbol{q}_{3i} &= \mathbf{0}.
\end{aligned}
\tag{39}
$$

Notice that we use $\mathbb{E}$ to denote the expectation on all randomness and use $\mathbb{E}_t$ to denote the expectation on the randomness at the $t$-th round, i.e., on the randomness of $\mathbf{X}_{(i),t}$ and $\boldsymbol{\epsilon}_{(i),t}$ for all $i \in [m]$. By Eqs. (38) and (39), we thus have

$$
\underset{t}{\mathbb{E}}\, \boldsymbol{\Delta}_t^{K=1} = \frac{1}{\sum_{i\in[m]} n_{(i),t}} \sum_{i\in[m]} \left(n_{(i),t}\left(1 - \alpha_{(i),t}\right) \boldsymbol{\Delta}_{t-1}^{K=1} + \alpha_{(i),t} n_{(i),t} \boldsymbol{\gamma}_{(i),t}\right).
\tag{40}
$$

Applying Eq. (40) recursively and recalling Eq. (10), we thus have

$$
\mathbb{E}[\boldsymbol{\Delta}_t^{K=1}] = \boldsymbol{g}_t^{K=1}.
\tag{41}
$$

By Assumption 1, we know that $\boldsymbol{\epsilon}_{(i),t}$ is independent of $\mathbf{X}_{(j),t}$ for all $i, j \in [m]$ and $\mathbb{E}\,\boldsymbol{\epsilon}_{(i),t} = \mathbf{0}$. Thus, we have

$$
\underset{t}{\mathbb{E}}[\boldsymbol{q}_{1i}^\top \boldsymbol{q}_{3j}] = \underset{t}{\mathbb{E}}[\boldsymbol{q}_{2i}^\top \boldsymbol{q}_{3j}] = 0.
$$

Thus, we have

$$
\begin{aligned}
\underset{t}{\mathbb{E}}\left\|\boldsymbol{\Delta}_t^{K=1}\right\|^2 &= \frac{1}{\left(\sum_{i\in[m]} n_{(i),t}\right)^2}\left(\sum_{i\in[m]}\left(\underset{t}{\mathbb{E}}\left\|\boldsymbol{q}_{1i}\right\|^2 + \underset{t}{\mathbb{E}}\left\|\boldsymbol{q}_{2i}\right\|^2 + \underset{t}{\mathbb{E}}\left\|\boldsymbol{q}_{3i}\right\|^2 + 2\underset{t}{\mathbb{E}}[\boldsymbol{q}_{1i}^\top \boldsymbol{q}_{2i}]\right) \right. \\
&\qquad \left. + \sum_{i\in[m]}\sum_{j\in[m]\setminus\{i\}}\left(\underset{t}{\mathbb{E}}[\boldsymbol{q}_{1i}^\top \boldsymbol{q}_{1j}] + \underset{t}{\mathbb{E}}[\boldsymbol{q}_{1i}^\top \boldsymbol{q}_{2j}] + \underset{t}{\mathbb{E}}[\boldsymbol{q}_{1j}^\top \boldsymbol{q}_{2i}] + \underset{t}{\mathbb{E}}[\boldsymbol{q}_{2i}^\top \boldsymbol{q}_{2j}]\right)\right) \\
&= \frac{1}{\left(\sum_{i\in[m]} n_{(i),t}\right)^2}\left(\sum_{i\in[m]}\left(\underset{t}{\mathbb{E}}\left\|\boldsymbol{q}_{1i}\right\|^2 + \underset{t}{\mathbb{E}}\left\|\boldsymbol{q}_{2i}\right\|^2 + \underset{t}{\mathbb{E}}\left\|\boldsymbol{q}_{3i}\right\|^2 + 2\underset{t}{\mathbb{E}}[\boldsymbol{q}_{1i}^\top \boldsymbol{q}_{2i}]\right) \right. \\
&\qquad \left. + \sum_{i\in[m]}\sum_{j\in[m]\setminus\{i\}}\left(\underset{t}{\mathbb{E}}[\boldsymbol{q}_{1i}^\top \boldsymbol{q}_{1j}] + 2\underset{t}{\mathbb{E}}[\boldsymbol{q}_{1i}^\top \boldsymbol{q}_{2j}] + \underset{t}{\mathbb{E}}[\boldsymbol{q}_{2i}^\top \boldsymbol{q}_{2j}]\right)\right) \\
&\quad \text{(since } \sum_{i\in[m]}\sum_{j\in[m]\setminus\{i\}} \boldsymbol{q}_{1i}^\top \boldsymbol{q}_{2j} + \boldsymbol{q}_{1j}^\top \boldsymbol{q}_{2i} = 2\sum_{i\in[m]}\sum_{j\in[m]\setminus\{i\}} \boldsymbol{q}_{1i}^\top \boldsymbol{q}_{2j}).
\end{aligned}
\tag{42}
$$

By Lemma 4, for any $i \in [m]$, we have

$$
\begin{aligned}
\underset{t}{\mathbb{E}}\left\|\boldsymbol{q}_{1i}\right\|^2 &= \left(n_{(i),t}^2 - 2\alpha_{(i),t} n_{(i),t}^2 + \alpha_{(i),t}^2 n_{(i),t}(n_{(i),t} + p + 1)\right)\left\|\boldsymbol{\Delta}_{t-1}^{K=1}\right\|^2 \\
&= \left(\left(1 - \alpha_{(i),t}\right)^2 n_{(i),t}^2 + \alpha_{(i),t}^2 n_{(i),t}(p+1)\right)\left\|\boldsymbol{\Delta}_{t-1}^{K=1}\right\|^2, \\
\underset{t}{\mathbb{E}}\left\|\boldsymbol{q}_{2i}\right\|^2 &= \alpha_{(i),t}^2 n_{(i),t}(n_{(i),t} + p + 1)\left\|\boldsymbol{\gamma}_{(i),t}\right\|^2, \\
\underset{t}{\mathbb{E}}\left\|\boldsymbol{q}_{3i}\right\|^2 &= \alpha_{(i),t}^2 p\, n_{(i),t} \sigma_{(i),t}^2, \\
\underset{t}{\mathbb{E}}[\boldsymbol{q}_{1i}^\top \boldsymbol{q}_{2i}] &= \left(\alpha_{(i),t} n_{(i),t}^2 - \alpha_{(i),t}^2 n_{(i),t}(n_{(i),t} + p + 1)\right)\boldsymbol{\Delta}_{t-1}^{K=1\top} \boldsymbol{\gamma}_{(i),t}.
\end{aligned}
\tag{43}
$$

Similarly, by Lemma 4, for any $i, j \in [m]$ where $i \neq j$, we have

$$
\begin{aligned}
\mathbb{E}[\boldsymbol{q}_{1i}^\top \boldsymbol{q}_{1j}] &= n_{(i),t} n_{(j),t}\left(1 - \alpha_{(i),t}\right)\left(1 - \alpha_{(j),t}\right)\left\|\boldsymbol{\Delta}_{t-1}^{K=1}\right\|^2, \\
\mathbb{E}[\boldsymbol{q}_{1i}^\top \boldsymbol{q}_{2j}] &= \left(\alpha_{(j),t} n_{(i),t} n_{(j),t} - \alpha_{(i),t}\alpha_{(j),t} n_{(i),t} n_{(j),t}\right)\boldsymbol{\Delta}_{t-1}^{K=1\top} \boldsymbol{\gamma}_{(j),t} \\
&= n_{(i),t} n_{(j),t} \alpha_{(j),t}\left(1 - \alpha_{(i),t}\right)\boldsymbol{\Delta}_{t-1}^{K=1\top} \boldsymbol{\gamma}_{(j),t}, \\
\mathbb{E}[\boldsymbol{q}_{2i}^\top \boldsymbol{q}_{2j}] &= \alpha_{(i),t}\alpha_{(j),t} n_{(i),t} n_{(j),t} \boldsymbol{\gamma}_{(i),t}^\top \boldsymbol{\gamma}_{(j),t}.
\end{aligned}
\tag{44}
$$

Plugging Eqs. (43) and (44) into Eq. (42), we thus have

$$
\mathbb{E}_t[\|\boldsymbol{\Delta}_t^{K=1}\|^2]
$$
$$
=\frac{\|\boldsymbol{\Delta}_{t-1}^{K=1}\|^2}{(\sum_{i\in[m]} n_{(i),t})^2}\left(\sum_{i\in[m]}\left((1-\alpha_{(i),t})^2 n_{(i),t}^2+\alpha_{(i),t}^2 n_{(i),t}(p+1)\right)+\sum_{i\in[m]}\sum_{j\in[m]\setminus\{j\}} n_{(i),t}n_{(j),t}(1-\alpha_{(i),t})(1-\alpha_{(j),t})\right)
$$
$$
+\frac{1}{(\sum_{i\in[m]} n_{(i),t})^2}\sum_{i\in[m]}\alpha_{(i),t}^2\left(pn_{(i),t}\sigma_{(i),t}^2+n_{(i),t}(n_{(i),t}+p+1)\|\boldsymbol{\gamma}_{(i),t}\|^2\right)
$$
$$
+2\frac{1}{(\sum_{i\in[m]} n_{(i),t})^2}\sum_{i\in[m]}\left(\alpha_{(i),t}n_{(i),t}^2-\alpha_{(i),t}^2 n_{(i),t}(n_{(i),t}+p+1)\right)\boldsymbol{\Delta}_{t-1}^{K=1\top}\boldsymbol{\gamma}_{(i),t}
$$
$$
+\frac{1}{(\sum_{i\in[m]} n_{(i),t})^2}\sum_{i\in[m]}\sum_{j\in[m]\setminus\{i\}}\left(2n_{(i),t}n_{(j),t}\alpha_{(j),t}\left(1-\alpha_{(i),t}\right)\boldsymbol{\Delta}_{t-1}^{K=1\top}\boldsymbol{\gamma}_{(j),t}+\alpha_{(i),t}\alpha_{(j),t}n_{(i),t}n_{(j),t}\boldsymbol{\gamma}_{(i),t}^\top\boldsymbol{\gamma}_{(j),t}\right).
$$

$$(45)$$

Notice that

$$
\left(\sum_{i\in[m]}\left((1-\alpha_{(i),t})^2 n_{(i),t}^2+\alpha_{(i),t}^2 n_{(i),t}(p+1)\right)+\sum_{i\in[m]}\sum_{j\in[m]\setminus\{j\}} n_{(i),t}n_{(j),t}(1-\alpha_{(i),t})(1-\alpha_{(j),t})\right)
$$
$$
=\frac{1}{(\sum_{i\in[m]} n_{(i),t})^2}\left(\sum_{i\in[m]} n_{(i),t}(1-\alpha_{(i),t})^2\right)^2+\frac{1}{(\sum_{i\in[m]} n_{(i),t})^2}\sum_{i\in[m]}\alpha_{(i),t}^2 n_{(i),t}(p+1)
$$
$$
=H_t \text{ (recalling Eq. (11))},
$$

and

$$
\frac{1}{(\sum_{i\in[m]} n_{(i),t})^2}\sum_{i\in[m]}\alpha_{(i),t}^2 n_{(i),t}(n_{(i),t}+p+1)\|\boldsymbol{\gamma}_{(i),t}\|^2
$$
$$
+\frac{1}{(\sum_{i\in[m]} n_{(i),t})^2}\sum_{i\in[m]}\sum_{j\in[m]\setminus\{i\}}\alpha_{(i),t}\alpha_{(j),t}n_{(i),t}n_{(j),t}\boldsymbol{\gamma}_{(i),t}^\top\boldsymbol{\gamma}_{(j),t}
$$
$$
=\frac{1}{(\sum_{i\in[m]} n_{(i),t})^2}\left\|\sum_{i\in[m]}\alpha_{(i),t}n_{(i),t}\boldsymbol{\gamma}_{(i),t}\right\|^2+\frac{1}{(\sum_{i\in[m]} n_{(i),t})^2}\sum_{i\in[m]}\alpha_{(i),t}^2 n_{(i),t}(p+1)\|\boldsymbol{\gamma}_{(i),t}\|^2,
$$

and

$$
2\frac{1}{(\sum_{i\in[m]} n_{(i),t})^2}\sum_{i\in[m]}\left(\alpha_{(i),t}n_{(i),t}^2-\alpha_{(i),t}^2 n_{(i),t}(n_{(i),t}+p+1)\right)\boldsymbol{\Delta}_{t-1}^{K=1\top}\boldsymbol{\gamma}_{(i),t}
$$
$$
+\frac{1}{(\sum_{i\in[m]} n_{(i),t})^2}\sum_{i\in[m]}\sum_{j\in[m]\setminus\{i\}}\left(2n_{(i),t}n_{(j),t}\alpha_{(j),t}\left(1-\alpha_{(i),t}\right)\boldsymbol{\Delta}_{t-1}^{K=1\top}\boldsymbol{\gamma}_{(j),t}\right)
$$
$$
=\frac{2}{(\sum_{i\in[m]} n_{(i),t})^2}\left(\sum_{i\in[m]} n_{(i),t}(1-\alpha_{(i),t})\right)\cdot\left(\sum_{i\in[m]} n_{(i),t}\alpha_{(i),t}\boldsymbol{\Delta}_{t-1}^{K=1\top}\boldsymbol{\gamma}_{(i),t}\right)
$$
$$
-\frac{2\sum_{i\in[m]}\alpha_{(i),t}^2 n_{(i),t}(p+1)\boldsymbol{\Delta}_{t-1}^{K=1\top}\boldsymbol{\gamma}_{(i),t}}{(\sum_{i\in[m]} n_{(i),t})^2}.
$$

Further, by Eq. (41) and recalling Eq. (12), we thus can rewrite Eq. (45) as

$$
\mathbb{E}\|\boldsymbol{\Delta}_t^{K=1}\|^2=H_t\,\mathbb{E}\|\boldsymbol{\Delta}_{t-1}^{K=1}\|^2+G_t. \tag{46}
$$

Applying Eq. (46) recursively, we thus have Eq. (13).

## C    PROOF OF THEOREM 2

Define

$$\boldsymbol{g}_l^{K<\infty} := \mathcal{F}\left(l, \boldsymbol{\Delta}_0, \text{seq}_t\left(\frac{\sum_{i\in[m]} n_{(i),t}(1-\alpha_{(i),t})^K}{\sum_{i\in[m]} n_{(i),t}}\right), \text{seq}_t\left(\frac{\sum_{i\in[m]} n_{(i),t}\left(1-(1-\alpha_{(i),t})^K\right)\boldsymbol{\gamma}_{(i),t}}{\sum_{i\in[m]} n_{(i),t}}\right)\right)$$

(47)

$$\mathcal{A}_{(i),t} := (1-\alpha_{(i),t})^2 + \frac{\alpha_{(i),t}^2(p+1)}{\tilde{n}_{(i),t}},$$

(48)

$$\mathcal{B}_{(i),t,k} := \frac{\alpha_{(i),t}^2 p\sigma_{(i),t}^2}{\tilde{n}_{(i),t}}$$

$$+ \left(\frac{\alpha_{(i),t}^2}{\tilde{n}_{(i),t}}(\tilde{n}_{(i),t}+p+1) + 2\alpha_{(i),t}\left(1-\frac{\alpha_{(i),t}}{\tilde{n}_{(i),t}}(\tilde{n}_{(i),t}+p+1)\right)\left(1-(1-\alpha_{(i),t})^{k-1}\right)\right)\left\|\boldsymbol{\gamma}_{(i),t}\right\|^2$$

$$+ 2\left(\alpha_{(i),t} - \frac{\alpha_{(i),t}^2}{\tilde{n}_{(i),t}}(\tilde{n}_{(i),t}+p+1)\right)(1-\alpha_{(i),t})^{k-1}\boldsymbol{\gamma}_{(i),t}^\top \boldsymbol{g}_{t-1}^{K<\infty},$$

(49)

$$\mathcal{J}_t := \frac{\sum_{i\in[m]} n_{(i),t}^2 \mathcal{A}_{(i),t}^K}{\left(\sum_{i\in[m]} n_{(i),t}\right)^2} + \frac{\sum_{i\in[m]}\sum_{j\in[m]\backslash\{i\}} n_{(i),t}n_{(j),t}(1-\alpha_{(i),t})^K(1-\alpha_{(j),t})^K}{\left(\sum_{i\in[m]} n_{(i),t}\right)^2},$$

(50)

$$\mathcal{Q}_t := \frac{\sum_{i\in[m]} n_{(i),t}^2 \sum_{k=1}^K \mathcal{B}_{(i),t,k}\mathcal{A}_{(i),t}^{K-k}}{\left(\sum_{i\in[m]} n_{(i),t}\right)^2}$$

$$+ \frac{1}{\left(\sum_{i\in[m]} n_{(i),t}\right)^2}\sum_{i\in[m]}\sum_{j\in[m]\backslash\{i\}} n_{(i),t}n_{(j),t}\left(2(1-\alpha_{(i),t})^K(1-(1-\alpha_{(j),t})^K)\boldsymbol{\gamma}_{(j),t}^\top\boldsymbol{g}_{t-1}^{K<\infty}\right.$$

$$\left. + (1-(1-\alpha_{(i),t})^K)(1-(1-\alpha_{(j),t})^K)\boldsymbol{\gamma}_{(i),t}^\top\boldsymbol{\gamma}_{(j),t}\right).$$

(51)

In the following, we use $\mathbb{E}_k$ to denote the expectation with respect to the randomness in the $k$-th batch.

We have

$$\boldsymbol{\Delta}_t^{K<\infty} = \boldsymbol{w}^* - \hat{\boldsymbol{w}}_{\text{avg},t}^{K<\infty}$$

$$= \boldsymbol{w}^* - \frac{1}{\sum_{i\in[m]} n_{(i),t}}\sum_{i\in[m]} n_{(i),t}\hat{\boldsymbol{w}}_{(i),t}$$

$$= \frac{1}{\sum_{i\in[m]} n_{(i),t}}\sum_{i\in[m]} n_{(i),t}(\boldsymbol{w}^* - \hat{\boldsymbol{w}}_{(i),t}) \text{ (since } \boldsymbol{w}^* = \frac{1}{\sum_{i\in[m]} n_{(i),t}}\sum_{i\in[m]} n_{(i),t}\boldsymbol{w}^*).$$

Thus, we have

$$\left\|\boldsymbol{\Delta}_t^{K<\infty}\right\|^2 = \frac{1}{\left(\sum_{i\in[m]} n_{(i),t}\right)^2}\sum_{i\in[m]} n_{(i),t}^2 \left\|\boldsymbol{w}^* - \hat{\boldsymbol{w}}_{(i),t}\right\|^2$$

$$+ \frac{1}{\left(\sum_{i\in[m]} n_{(i),t}\right)^2}\sum_{i\in[m]}\sum_{j\in[m]\backslash\{i\}} n_{(i),t}n_{(j),t}(\boldsymbol{w}^* - \hat{\boldsymbol{w}}_{(i),t})^\top(\boldsymbol{w}^* - \hat{\boldsymbol{w}}_{(j),t}).$$

(52)

By Assumption 1, we know that at round $t$, different agents' data are independent with each other. Thus, we have

$$\mathbb{E}_t(\boldsymbol{w}^* - \hat{\boldsymbol{w}}_{(i),t})^\top(\boldsymbol{w}^* - \hat{\boldsymbol{w}}_{(j),t}) = \mathbb{E}_t(\boldsymbol{w}^* - \hat{\boldsymbol{w}}_{(i),t})^\top \mathbb{E}_t(\boldsymbol{w}^* - \hat{\boldsymbol{w}}_{(j),t}).$$

Thus, by Eq. (52), to calculate $\mathbb{E}_t\left\|\boldsymbol{\Delta}_t^{K<\infty}\right\|^2$, it remains to calculate $\mathbb{E}_t\left\|\boldsymbol{w}^* - \hat{\boldsymbol{w}}_{(i),t}\right\|^2$ and $\mathbb{E}_t(\boldsymbol{w}^* - \hat{\boldsymbol{w}}_{(i),t})$ for all $i \in [m]$. To that end, we have

$$\hat{\boldsymbol{w}}_{(i),t,k} = \left(\mathbf{I}_p - \frac{\alpha_{(i),t}}{\tilde{n}_{(i),t}}\mathbf{X}_{(i),t,k}\mathbf{X}_{(i),t,k}^\top\right)\hat{\boldsymbol{w}}_{(i),t,k-1} + \frac{\alpha_{(i),t}}{\tilde{n}_{(i),t}}\mathbf{X}_{(i),t,k}(\mathbf{X}_{(i),t,k}^\top\boldsymbol{w}_{(i),t} + \boldsymbol{\epsilon}_{(i),t,k}).$$

We thus have

$$
\boldsymbol{w}^* - \hat{\boldsymbol{w}}_{(i),t,k} = \left( \mathbf{I}_p - \frac{\alpha_{(i),t}}{\tilde{n}_{(i),t}} \mathbf{X}_{(i),t,k} \mathbf{X}_{(i),t,k}^\top \right) (\boldsymbol{w}^* - \hat{\boldsymbol{w}}_{(i),t,k-1}) + \frac{\alpha_{(i),t}}{\tilde{n}_{(i),t}} \mathbf{X}_{(i),t,k} \mathbf{X}_{(i),t,k}^\top (\boldsymbol{w}^* - \boldsymbol{w}_{(i),t})
$$
$$
+ \frac{\alpha_{(i),t}}{\tilde{n}_{(i),t}} \mathbf{X}_{(i),t,k} \boldsymbol{\epsilon}_{(i),t,k}.
$$
(53)

By Lemma 4 and recalling Eq. (3), we thus have

$$
\mathbb{E}_k (\boldsymbol{w}^* - \hat{\boldsymbol{w}}_{(i),t,k}) = (1 - \alpha_{(i),t})(\boldsymbol{w}^* - \hat{\boldsymbol{w}}_{(i),t,k-1}) + \alpha_{(i),t} \boldsymbol{\gamma}_{(i),t}.
$$
(54)

Applying Eq. (54) recursively and recalling that $\hat{\boldsymbol{w}}_{(i),t,0} = \boldsymbol{\Delta}_{t-1}^{K<\infty}$, we thus have

$$
\mathbb{E}_{1,2,\cdots,k} (\boldsymbol{w}^* - \hat{\boldsymbol{w}}_{(i),t,k}) = (1 - \alpha_{(i),t})^k \boldsymbol{\Delta}_{t-1}^{K<\infty} + \left( 1 - (1 - \alpha_{(i),t})^k \right) \boldsymbol{\gamma}_{(i),t}.
$$
(55)

By letting $k = K$ in Eq. (55) and $\hat{\boldsymbol{w}}_{(i),t,K} = \hat{\boldsymbol{w}}_{(i),t}$, we thus have

$$
\mathbb{E}_t (\boldsymbol{w}^* - \hat{\boldsymbol{w}}_{(i),t}) = (1 - \alpha_{(i),t})^K \boldsymbol{\Delta}_{t-1}^{K<\infty} + \left( 1 - (1 - \alpha_{(i),t})^K \right) \boldsymbol{\gamma}_{(i),t}.
$$
(56)

Plugging Eq. (56) into Eq. (52), we thus have

$$
\mathbb{E}_t \left\| \boldsymbol{\Delta}_t^{K<\infty} \right\|^2 = \frac{1}{(\sum_{i \in [m]} n_{(i),t})^2} \sum_{i \in [m]} n_{(i),t}^2 \mathbb{E}_t \left\| \boldsymbol{w}^* - \hat{\boldsymbol{w}}_{(i),t} \right\|^2
$$
$$
+ \frac{1}{(\sum_{i \in [m]} n_{(i),t})^2} \sum_{i \in [m]} \sum_{j \in [m] \setminus \{i\}} n_{(i),t} n_{(j),t} \mathbb{E}_t (\boldsymbol{w}^* - \hat{\boldsymbol{w}}_{(i),t})^\top \mathbb{E}_t (\boldsymbol{w}^* - \hat{\boldsymbol{w}}_{(j),t})
$$
(57)

$$
= \frac{1}{(\sum_{i \in [m]} n_{(i),t})^2} \sum_{i \in [m]} n_{(i),t}^2 \mathbb{E}_t \left\| \boldsymbol{w}^* - \hat{\boldsymbol{w}}_{(i),t} \right\|^2
$$
$$
+ \frac{1}{(\sum_{i \in [m]} n_{(i),t})^2} \sum_{i \in [m]} \sum_{j \in [m] \setminus \{i\}} n_{(i),t} n_{(j),t} \left( (1 - \alpha_{(i),t})^K (1 - \alpha_{(j),t})^K \left\| \boldsymbol{\Delta}_{t-1}^{K<\infty} \right\|^2 \right.
$$
$$
+ (1 - \alpha_{(i),t})^K (1 - (1 - \alpha_{(j),t})^K) \boldsymbol{\gamma}_{(j),t}^\top \boldsymbol{\Delta}_{t-1}^{K<\infty} + (1 - \alpha_{(j),t})^K (1 - (1 - \alpha_{(i),t})^K) \boldsymbol{\gamma}_{(i),t}^\top \boldsymbol{\Delta}_{t-1}^{K<\infty}
$$
$$
\left. + (1 - (1 - \alpha_{(i),t})^K)(1 - (1 - \alpha_{(j),t})^K) \boldsymbol{\gamma}_{(i),t}^\top \boldsymbol{\gamma}_{(j),t} \right)
$$
$$
= \frac{1}{(\sum_{i \in [m]} n_{(i),t})^2} \sum_{i \in [m]} n_{(i),t}^2 \mathbb{E}_t \left\| \boldsymbol{w}^* - \hat{\boldsymbol{w}}_{(i),t} \right\|^2
$$
$$
+ \frac{1}{(\sum_{i \in [m]} n_{(i),t})^2} \sum_{i \in [m]} \sum_{j \in [m] \setminus \{i\}} n_{(i),t} n_{(j),t} \left( (1 - \alpha_{(i),t})^K (1 - \alpha_{(j),t})^K \left\| \boldsymbol{\Delta}_{t-1}^{K<\infty} \right\|^2 \right.
$$
$$
+ 2(1 - \alpha_{(i),t})^K (1 - (1 - \alpha_{(j),t})^K) \boldsymbol{\gamma}_{(j),t}^\top \boldsymbol{\Delta}_{t-1}^{K<\infty}
$$
$$
\left. + (1 - (1 - \alpha_{(i),t})^K)(1 - (1 - \alpha_{(j),t})^K) \boldsymbol{\gamma}_{(i),t}^\top \boldsymbol{\gamma}_{(j),t} \right).
$$
(58)

Notice that in Eq. (57) we use $\mathbb{E}_t (\boldsymbol{w}^* - \hat{\boldsymbol{w}}_{(i),t})^\top (\boldsymbol{w}^* - \hat{\boldsymbol{w}}_{(j),t}) = \mathbb{E}_t (\boldsymbol{w}^* - \hat{\boldsymbol{w}}_{(i),t})^\top \mathbb{E}_t (\boldsymbol{w}^* - \hat{\boldsymbol{w}}_{(j),t})$ for $i \neq j$, since $\hat{\boldsymbol{w}}_{(i),t}$ and $\hat{\boldsymbol{w}}_{(j),t}$ are independent with respect to the randomness during the local updates at round $t$.

By Eqs. (5) and (56), we thus have

$$
\mathbb{E} \, \boldsymbol{\Delta}_t^{K<\infty} = \frac{\sum_{i \in [m]} n_{(i),t} (1 - \alpha_{(i),t})^K}{\sum_{i \in [m]} n_{(i),t}} \mathbb{E} \, \boldsymbol{\Delta}_{t-1}^{K<\infty} + \frac{\sum_{i \in [m]} n_{(i),t} \left( 1 - (1 - \alpha_{(i),t})^K \right) \boldsymbol{\gamma}_{(i),t}}{\sum_{i \in [m]} n_{(i),t}}.
$$
(59)

Applying Eq. (59) recursively and recalling Eq. (8), we thus have

$$
\mathbb{E}[\boldsymbol{\Delta}_l^{K<\infty}] = \boldsymbol{g}_l^{K<\infty},
$$
(60)

where $\boldsymbol{g}_l^{K<\infty}$ is defined in Eq. (47).

By Eq. (53), we have

$$
\underset{k}{\mathbb{E}} \left\| \boldsymbol{w}^* - \hat{\boldsymbol{w}}_{(i),t,k} \right\|^2
$$

$$
= (\boldsymbol{w}^* - \hat{\boldsymbol{w}}_{(i),t,k-1})^\top \left( \mathbf{I}_p - 2 \frac{\alpha_{(i),t}}{\tilde{n}_{(i),t}} \mathbf{X}_{(i),t,k} \mathbf{X}_{(i),t,k}^\top + \frac{\alpha_{(i),t}^2}{\tilde{n}_{(i),t}^2} \mathbf{X}_{(i),t,k} \mathbf{X}_{(i),t,k}^\top \mathbf{X}_{(i),t,k} \mathbf{X}_{(i),t,k}^\top \right) (\boldsymbol{w}^* - \hat{\boldsymbol{w}}_{(i),t,k-1})
$$

$$
+ \boldsymbol{\gamma}_{(i),t}^\top \frac{\alpha_{(i),t}^2}{\tilde{n}_{(i),t}^2} \mathbf{X}_{(i),t,k} \mathbf{X}_{(i),t,k}^\top \mathbf{X}_{(i),t,k} \mathbf{X}_{(i),t,k}^\top \boldsymbol{\gamma}_{(i),t} + \boldsymbol{\epsilon}_{(i),t,k}^\top \frac{\alpha_{(i),t}^2}{\tilde{n}_{(i),t}^2} \mathbf{X}_{(i),t,k}^\top \mathbf{X}_{(i),t,k} \boldsymbol{\epsilon}_{(i),t,k}
$$

$$
+ 2 \frac{\alpha_{(i),t}}{\tilde{n}_{(i),t}} \boldsymbol{\gamma}_{(i),t}^\top \mathbf{X}_{(i),t,k} \mathbf{X}_{(i),t,k}^\top \left( \mathbf{I}_p - \frac{\alpha_{(i),t}}{\tilde{n}_{(i),t}} \mathbf{X}_{(i),t,k} \mathbf{X}_{(i),t,k}^\top \right) (\boldsymbol{w}^* - \hat{\boldsymbol{w}}_{(i),t,k-1})
$$

$$
= \left( 1 - 2\alpha_{(i),t} + \frac{\alpha_{(i),t}^2}{\tilde{n}_{(i),t}} (\tilde{n}_{(i),t} + p + 1) \right) \left\| \boldsymbol{w}^* - \hat{\boldsymbol{w}}_{(i),t,k-1} \right\|^2 + \frac{\alpha_{(i),t}^2}{\tilde{n}_{(i),t}} (\tilde{n}_{(i),t} + p + 1) \left\| \boldsymbol{\gamma}_{(i),t} \right\|^2
$$

$$
+ \alpha_{(i),t}^2 \frac{p}{\tilde{n}_{(i),t}} \sigma_{(i),t}^2 + 2\alpha_{(i),t} \left( 1 - \frac{\alpha_{(i),t}}{\tilde{n}_{(i),t}} (\tilde{n}_{(i),t} + p + 1) \right) \boldsymbol{\gamma}_{(i),t}^\top (\boldsymbol{w}^* - \hat{\boldsymbol{w}}_{(i),t,k-1}) \quad \text{(by Lemma 4)}.
$$

$$
\tag{61}
$$

Plugging Eq. (55) into Eq. (61), we have

$$
\underset{1,2,\cdots,k}{\mathbb{E}} \left\| \boldsymbol{w}^* - \hat{\boldsymbol{w}}_{(i),t,k} \right\|^2
$$

$$
= \left( (1 - \alpha_{(i),t})^2 + \frac{\alpha_{(i),t}^2 (p+1)}{\tilde{n}_{(i),t}} \right) \underset{1,2,\cdots,k-1}{\mathbb{E}} \left\| \boldsymbol{w}^* - \hat{\boldsymbol{w}}_{(i),t,k-1} \right\|^2 + \frac{\alpha_{(i),t}^2}{\tilde{n}_{(i),t}} (\tilde{n}_{(i),t} + p + 1) \left\| \boldsymbol{\gamma}_{(i),t} \right\|^2
$$

$$
+ \alpha_{(i),t}^2 \frac{p}{\tilde{n}_{(i),t}} \sigma_{(i),t}^2 + 2\alpha_{(i),t} \left( 1 - \frac{\alpha_{(i),t}}{\tilde{n}_{(i),t}} (\tilde{n}_{(i),t} + p + 1) \right) (1 - \alpha_{(i),t})^{k-1} \boldsymbol{\gamma}_{(i),t}^\top \boldsymbol{\Delta}_{t-1}^{K<\infty}
$$

$$
+ 2\alpha_{(i),t} \left( 1 - \frac{\alpha_{(i),t}}{\tilde{n}_{(i),t}} (\tilde{n}_{(i),t} + p + 1) \right) \left( 1 - (1 - \alpha_{(i),t})^{k-1} \right) \left\| \boldsymbol{\gamma}_{(i),t} \right\|^2
$$

$$
= \mathcal{A}_{(i),t} \, \mathbb{E} \left\| \boldsymbol{w}^* - \hat{\boldsymbol{w}}_{(i),t,k-1} \right\|^2 + \mathcal{B}'_{(i),t,k}, \tag{62}
$$

where $\mathcal{A}_{(i),t}$ is defined in Eq. (48) and

$$
\mathcal{B}'_{(i),t,k}
$$

$$
:= \frac{\alpha_{(i),t}^2 p \sigma_{(i),t}^2}{\tilde{n}_{(i),t}}
$$

$$
+ \left( \frac{\alpha_{(i),t}^2}{\tilde{n}_{(i),t}} (\tilde{n}_{(i),t} + p + 1) + 2\alpha_{(i),t} \left( 1 - \frac{\alpha_{(i),t}}{\tilde{n}_{(i),t}} (\tilde{n}_{(i),t} + p + 1) \right) \left( 1 - (1 - \alpha_{(i),t})^{k-1} \right) \right) \left\| \boldsymbol{\gamma}_{(i),t} \right\|^2
$$

$$
+ 2 \left( \alpha_{(i),t} - \frac{\alpha_{(i),t}^2}{\tilde{n}_{(i),t}} (\tilde{n}_{(i),t} + p + 1) \right) (1 - \alpha_{(i),t})^{k-1} \boldsymbol{\gamma}_{(i),t}^\top \boldsymbol{\Delta}_{t-1}^{K<\infty}.
$$

We also define $\mathcal{B}_{(i),t,k}$ by replacing $\boldsymbol{\Delta}_{t-1}^{K<\infty}$ in $\mathcal{B}'_{(i),t,k}$ with $\mathcal{F}_{t-1}$, i.e., Eq. (49).

Applying Eq. (62) recursively over $k = 1, 2, \cdots, K$, we thus have

$$
\underset{t}{\mathbb{E}} \left\| \boldsymbol{w}^* - \hat{\boldsymbol{w}}_{(i),t} \right\|^2 = \mathcal{A}_{(i),t}^K \left\| \boldsymbol{\Delta}_{t-1}^{K<\infty} \right\|^2 + \sum_{k=1}^K \mathcal{B}_{(i),t,k} \mathcal{A}_{(i),t}^{K-k}. \tag{63}
$$

Plugging Eqs. (60) and (63) into Eq. (58), we thus have

$$
\mathbb{E} \left\| \boldsymbol{\Delta}_t^{K<\infty} \right\|^2 = \mathcal{J}_t \, \mathbb{E} \left\| \boldsymbol{\Delta}_{t-1}^{K<\infty} \right\|^2 + \mathcal{Q}_t, \tag{64}
$$

where $\mathcal{J}_t$ is defined in Eq. (50) and $\mathcal{Q}_t$ is defined in Eq. (51).

Applying Eq. (64) recursively, we thus have Eq. (15).

# D  PROOF OF PROPOSITION 1

*Proof.* (1) Since $\tilde{n}$ is fixed, then $\mathcal{A}$ does not change with $K$. When $t \to \infty$, the value of Eq. (16) becomes

$$\frac{1}{1-\mathcal{J}} \frac{\alpha^2 p \sigma^2}{mn} \cdot \frac{1 - \mathcal{A}^K}{1 - \mathcal{A}}. \tag{65}$$

The only component related to $K$ in Eq. (65) is $\frac{1-\mathcal{A}^K}{1-\mathcal{J}}$, thus $K_{\text{opt}} = \arg\min_K \frac{1-\mathcal{A}^K}{1-\mathcal{J}}$. Notice that for any finite $K$, we must have

$$\mathcal{A}^K = \left( (1-\alpha)^2 + \frac{\alpha^2(p+1)}{\tilde{n}} \right)^K > (1-\alpha)^{2K}.$$

Thus, we have

$$\mathcal{J} = \frac{1}{m} \mathcal{A}^K + \frac{m-1}{m}(1-\alpha)^{2K} < \mathcal{A}^K,$$

which implies that $\frac{1-\mathcal{A}^K}{1-\mathcal{J}} < 1$ for any finite $K$. Meanwhile, $\lim_{K \to \infty} \frac{1-\mathcal{A}^K}{1-\mathcal{J}} = 1$. Thus, $K_{\text{opt}}$ should be finite.

(2) Since $\tilde{n}$ is fixed, then $\mathcal{A}$ does not change with $K$. When $\sigma = 0$, Eq. (16) becomes $\mathcal{J}^t \|\mathbf{\Delta}_0\|^2$. Notice that $\mathcal{J}$ is strictly monotone decreasing w.r.t. $K$. Therefore, $K_{\text{opt}} = \infty$.

(3) Since we use $n/K$ to replace $\lfloor n/K \rfloor$, we have $K_{\text{opt}} = \arg\min_K f(K)$ where

$$f(K) := \left( (1-\alpha)^2 + K\frac{\alpha^2(p+1)}{n} \right)^K + (m-1)(1-\alpha)^{2K}.$$

Calculating the derivative, we have

$$\begin{aligned}
\frac{\partial f(K)}{\partial K} =& \frac{\alpha^2(p+1)}{n} \left( (1-\alpha)^2 + K\frac{\alpha^2(p+1)}{n} \right)^K \ln\left( (1-\alpha)^2 + K\frac{\alpha^2(p+1)}{n} \right) \\
&+ (m-1)(1-\alpha)^{2K} \ln((1-\alpha)^2).
\end{aligned} \tag{66}$$

When $\left( (1-\alpha)^2 + K\frac{\alpha^2(p+1)}{n} \right) < 1$, we have $\frac{\partial f(K)}{\partial K} < 0$.

For any $\delta > 0$, when

$$\left( (1-\alpha)^2 + K\frac{\alpha^2(p+1)}{n} \right) > 1 + \delta,$$

$$\frac{\alpha^2(p+1)}{n}(1 + K\delta)\ln(1+\delta) > (m-1)\ln\frac{1}{(1-\alpha)^2},$$

we have Eq. (66) $> 0$. (Notice that we utilize the face that $(1-\alpha)^{2K} < 1$ and $(1+\delta)^K \geq 1 + K\delta$.) Solving those inequalities by further letting $\ln(1+\delta) = \ln\frac{1}{(1-\alpha)^2}$, we thus have

$$\frac{n}{(p+1)}\left( \frac{2}{\alpha} - 1 \right) \leq K_{\text{opt}} \leq \frac{n}{\alpha^2(p+1)} \cdot \max\left\{ (2\alpha - \alpha^2)\left( 1 + \frac{1}{(1-\alpha)^2} \right), (m-2)\frac{(1-\alpha)^2}{2\alpha - \alpha^2} \right\}.$$

When $\alpha \leq 0.1$ and $m \geq 3$, we can further relax the above inequality as

$$\frac{n}{p+1}\left( \frac{2}{\alpha} - 1 \right) \leq K_{\text{opt}} \leq \frac{n}{p+1}\frac{(m-2)}{\alpha^3}.$$

$\square$

# E  PROOF OF THEOREM 3

*Proof.* In the overparameterized situation, after each agent trains to converge, we have

$$\hat{\boldsymbol{w}}_{(i),t}^{K=\infty} = \mathbf{X}_{(i),t} \left( \mathbf{X}_{(i),t}^{\top} \mathbf{X}_{(i),t} \right)^{-1} \left( \boldsymbol{y}_{(i),t} - \mathbf{X}_{(i),t}^{\top} \hat{\boldsymbol{w}}_{\text{avg},t-1}^{K=\infty} \right) + \hat{\boldsymbol{w}}_{\text{avg},t-1}^{K=\infty}. \tag{67}$$

For any $i \in [m]$, we define $\mathbf{P}_{(i),t} \in \mathbb{R}^{p \times p}$ as

$$\mathbf{P}_{(i),t} \coloneqq \mathbf{X}_{(i),t} \left( \mathbf{X}_{(i),t}^{\top} \mathbf{X}_{(i),t} \right)^{-1} \mathbf{X}_{(i),t}^{\top}. \tag{68}$$

(We know $\mathbf{P}_{(i),t}$ is an orthogonal projection since $\mathbf{P}_{(i),t}\mathbf{P}_{(i),t} = \mathbf{P}_{(i),t}$ and $\mathbf{P}_{(i),t}^{\top} = \mathbf{P}_{(i),t}$.) By Eqs. (2), (67) and (68), we thus have

$$\hat{\boldsymbol{w}}_{(i),t}^{K=\infty} = \mathbf{P}_{(i),t}\boldsymbol{w}_{(i),t} + (\mathbf{I}_p - \mathbf{P}_{(i),t})\hat{\boldsymbol{w}}_{\text{avg},t-1}^{K=\infty} + \mathbf{X}_{(i),t} \left( \mathbf{X}_{(i),t}^{\top} \mathbf{X}_{(i),t} \right)^{-1} \boldsymbol{\epsilon}_{(i),t}. \tag{69}$$

We thus have

$$\boldsymbol{\Delta}_t^{K=\infty}$$
$$= \boldsymbol{w}^* - \hat{\boldsymbol{w}}_{\text{avg},t}^{K=\infty} \quad \text{(by Eq. (7))}$$
$$= \boldsymbol{w}^* - \frac{1}{\sum_{i \in [m]} n_{(i),t}} \sum_{i \in [m]} n_{(i),t} \left( \mathbf{P}_{(i),t}\boldsymbol{w}_{(i),t} + (\mathbf{I}_p - \mathbf{P}_{(i),t})\hat{\boldsymbol{w}}_{\text{avg},t-1}^{K=\infty} + \mathbf{X}_{(i),t} \left( \mathbf{X}_{(i),t}^{\top} \mathbf{X}_{(i),t} \right)^{-1} \boldsymbol{\epsilon}_{(i),t} \right)$$
$$\quad \text{(by Eqs. (5) and (69))}$$
$$= \frac{1}{\sum_{i \in [m]} n_{(i),t}} \sum_{i \in [m]} n_{(i),t} \left( \mathbf{P}_{(i),t}(\boldsymbol{w}^* - \boldsymbol{w}_{(i),t}) + (\mathbf{I}_p - \mathbf{P}_{(i),t})(\boldsymbol{w}^* - \hat{\boldsymbol{w}}_{\text{avg},t-1}^{K=\infty}) - \mathbf{X}_{(i),t} \left( \mathbf{X}_{(i),t}^{\top} \mathbf{X}_{(i),t} \right)^{-1} \boldsymbol{\epsilon}_{(i),t} \right)$$
$$\quad \left(\text{since } \boldsymbol{w}^* = \frac{\sum_{i \in [m]} n_{(i),t}(\mathbf{P}_{(i),t} + \mathbf{I}_p - \mathbf{P}_{(i),t})\boldsymbol{w}^*}{\sum_{i \in [m]} n_{(i),t}}\right)$$
$$= \frac{1}{\sum_{i \in [m]} n_{(i),t}} \sum_{i \in [m]} n_{(i),t} \left( \mathbf{P}_{(i),t}\boldsymbol{\gamma}_{(i),t} + (\mathbf{I}_p - \mathbf{P}_{(i),t})\boldsymbol{\Delta}_{t-1}^{K=\infty} - \mathbf{X}_{(i),t} \left( \mathbf{X}_{(i),t}^{\top} \mathbf{X}_{(i),t} \right)^{-1} \boldsymbol{\epsilon}_{(i),t} \right)$$
$$\quad \text{(by Eqs. (3) and (7))}. \tag{70}$$

For any $i, j \in [m]$, because $\boldsymbol{\epsilon}_{(j),t}$ is independent of $\boldsymbol{\Delta}_{t-1}^{K=\infty}$ and $\mathbf{X}_{(i),t}$, and also because $\boldsymbol{\epsilon}_{(j),t}$ has zero mean (by Assumption 1), we have

$$\mathbb{E}\left[ \left(\mathbf{P}_{(i),t}\boldsymbol{\gamma}_{(i),t}\right)^{\top} \mathbf{X}_{(j),t} \left( \mathbf{X}_{(j),t}^{\top} \mathbf{X}_{(j),t} \right)^{-1} \boldsymbol{\epsilon}_{(j),t} \right]$$
$$= \mathbb{E}\left[ \left((\mathbf{I}_p - \mathbf{P}_{(i),t})\boldsymbol{\Delta}_{t-1}^{K=\infty}\right)^{\top} \mathbf{X}_{(i),t} \left( \mathbf{X}_{(i),t}^{\top} \mathbf{X}_{(i),t} \right)^{-1} \boldsymbol{\epsilon}_{(i),t} \right]$$
$$= 0, \tag{71}$$

and

$$\mathbb{E}\left[ \mathbf{X}_{(i),t} \left( \mathbf{X}_{(i),t}^{\top} \mathbf{X}_{(i),t} \right)^{-1} \boldsymbol{\epsilon}_{(i),t} \right] = \mathbf{0}. \tag{72}$$

Since $\mathbf{P}_{(i),t}(\mathbf{I}_p - \mathbf{P}_{(i),t}) = \mathbf{0}$, we have

$$\left(\mathbf{P}_{(i),t}\boldsymbol{\gamma}_{(i),t}\right)^{\top} (\mathbf{I}_p - \mathbf{P}_{(i),t})\boldsymbol{\Delta}_{t-1}^{K=\infty} = 0. \tag{73}$$

Thus, by Eqs. (70), (71) and (73), we have

$$
\mathbb{E}_t \left\| \mathbf{\Delta}_t^{K=\infty} \right\|^2
$$

$$
= \frac{\sum_{i \in [m]} n_{(i),t}^2 \left( \mathbb{E}_t \left\| (\mathbf{I}_p - \mathbf{P}_{(i),t}) \mathbf{\Delta}_{t-1}^{K=\infty} \right\|^2 + \mathbb{E}_t \left\| \mathbf{P}_{(i),t} \boldsymbol{\gamma}_{(i),t} \right\|^2 + \mathbb{E}_t \left\| \mathbf{X}_{(i),t} \left( \mathbf{X}_{(i),t}^\top \mathbf{X}_{(i),t} \right)^{-1} \boldsymbol{\epsilon}_{(i),t} \right\|^2 \right)}{\left( \sum_{i \in [m]} n_{(i),t} \right)^2}
$$

$$
+ \frac{1}{\left( \sum_{i \in [m]} n_{(i),t} \right)^2} \sum_{i \in [m]} \sum_{j \in [m] \setminus \{i\}} n_{(i),t} n_{(j),t} \left( \boldsymbol{\gamma}_{(j),t}^\top \mathbf{P}_{(j),t} \mathbf{P}_{(i),t} \boldsymbol{\gamma}_{(i),t} \right.
$$

$$
\left. + \mathbf{\Delta}_{t-1}^{K=\infty \top} (\mathbf{I}_p - \mathbf{P}_{(j),t})(\mathbf{I}_p - \mathbf{P}_{(i),t}) \mathbf{\Delta}_{t-1}^{K=\infty} + 2 \boldsymbol{\gamma}_{(j),t}^\top \mathbf{P}_{(j),t} (\mathbf{I}_p - \mathbf{P}_{(i),t}) \mathbf{\Delta}_{t-1}^{K=\infty} \right). \tag{74}
$$

For any $i \in [m]$, we have

$$
\mathbb{E}_t \left\| \mathbf{P}_{(i),t} \boldsymbol{\gamma}_{(i),t} \right\|^2 = \frac{n_{(i),t}}{p} \left\| \boldsymbol{\gamma}_{(i),t} \right\|^2 \quad \text{(by Lemma 2),} \tag{75}
$$

$$
\mathbb{E}_t \left\| (\mathbf{I}_p - \mathbf{P}_{(i),t}) \mathbf{\Delta}_{t-1}^{K=\infty} \right\|^2 = \left( 1 - \frac{n_{(i),t}}{p} \right) \left\| \mathbf{\Delta}_{t-1}^{K=\infty} \right\|^2 \quad \text{(by Lemma 2),} \tag{76}
$$

$$
\mathbb{E}_t \left\| \mathbf{X}_{(i),t} \left( \mathbf{X}_{(i),t}^\top \mathbf{X}_{(i),t} \right)^{-1} \boldsymbol{\epsilon}_{(i),t} \right\|^2 = \frac{n_{(i),t} \sigma_i^2}{p - n_{(i),t} - 1} \quad \text{(by Lemma 3).} \tag{77}
$$

For any $i, j \in [m]$ where $i \neq j$, we have

$$
\mathbb{E}_t \left[ \mathbf{\Delta}_{t-1}^{K=\infty \top} (\mathbf{I}_p - \mathbf{P}_{(j),t})(\mathbf{I}_p - \mathbf{P}_{(i),t}) \mathbf{\Delta}_{t-1}^{K=\infty} \right]
$$

$$
= \mathbb{E}_t \left[ (\mathbf{I}_p - \mathbf{P}_{(i),t}) \mathbf{\Delta}_{t-1}^{K=\infty} \right]^\top \mathbb{E}_t \left[ (\mathbf{I}_p - \mathbf{P}_{(j),t}) \mathbf{\Delta}_{t-1}^{K=\infty} \right]
$$

$$
\text{(since } \mathbf{P}_{(i),t} \text{ and } \mathbf{P}_{(j),t} \text{ are independent when } i \neq j\text{)}
$$

$$
= \left( 1 - \frac{n_{(i),t}}{p} \right) \left( 1 - \frac{n_{(j),t}}{p} \right) \left\| \mathbf{\Delta}_{t-1}^{K=\infty} \right\|^2 \quad \text{(by Lemma 5).} \tag{78}
$$

Similarly, for $i \neq j$, we have

$$
\mathbb{E}_t \left[ \boldsymbol{\gamma}_{(j),t}^\top \mathbf{P}_{(j),t} \mathbf{P}_{(i),t} \boldsymbol{\gamma}_{(i),t} \right] = \frac{n_{(i),t} n_{(j),t}}{p^2} \boldsymbol{\gamma}_{(j),t}^\top \boldsymbol{\gamma}_{(i),t} \quad \text{(by Lemma 5),} \tag{79}
$$

and

$$
\mathbb{E}_t \left[ \boldsymbol{\gamma}_{(j),t}^\top \mathbf{P}_{(j),t} (\mathbf{I}_p - \mathbf{P}_{(i),t}) \mathbf{\Delta}_{t-1}^{K=\infty} \right] = \frac{n_{(j),t}}{p} \left( 1 - \frac{n_{(i),t}}{p} \right) \boldsymbol{\gamma}_{(j),t}^\top \mathbf{\Delta}_{t-1}^{K=\infty} \quad \text{(by Lemma 5).} \tag{80}
$$

Plugging Eqs. (78) to (80) and (75) to (77) into Eq. (74), we thus have

$$
\mathbb{E}_t \left\| \mathbf{\Delta}_t^{K=\infty} \right\|^2
$$

$$
= \frac{\sum_{i \in [m]} n_{(i),t}^2 \left( \left( 1 - \frac{n_{(i),t}}{p} \right) \left\| \mathbf{\Delta}_{t-1}^{K=\infty} \right\|^2 + \frac{n_{(i),t}}{p} \left\| \boldsymbol{\gamma}_{(i),t} \right\|^2 + \frac{n_{(i),t} \sigma_{(i),t}^2}{p - n_{(i),t} - 1} \right)}{\left( \sum_{i \in [m]} n_{(i),t} \right)^2}
$$

$$
+ \frac{1}{\left( \sum_{i \in [m]} n_{(i),t} \right)^2} \sum_{i \in [m]} \sum_{j \in [m] \setminus \{i\}} n_{(i),t} n_{(j),t} \left( \frac{n_{(i),t} n_{(j),t}}{p^2} \boldsymbol{\gamma}_{(j),t}^\top \boldsymbol{\gamma}_{(i),t} \right.
$$

$$
\left. + \left( 1 - \frac{n_{(i),t}}{p} \right) \left( 1 - \frac{n_{(j),t}}{p} \right) \left\| \mathbf{\Delta}_{t-1}^{K=\infty} \right\|^2 + 2 \frac{n_{(j),t}}{p} \left( 1 - \frac{n_{(i),t}}{p} \right) \boldsymbol{\gamma}_{(j),t}^\top \mathbf{\Delta}_{t-1}^{K=\infty} \right). \tag{81}
$$

By Eq. (70), we also have

$$
\mathbb{E}_t[\mathbf{\Delta}_t^{K=\infty}] = \frac{1}{\sum_{i \in [m]} n_{(i),t}} \sum_{i \in [m]} n_{(i),t} \left( \frac{n_{(i),t}}{p} \boldsymbol{\gamma}_{(i),t} + \left( 1 - \frac{n_{(i),t}}{p} \right) \mathbf{\Delta}_{t-1}^{K=\infty} \right). \tag{82}
$$

Applying Eq. (82) recursively, we thus have

$$\mathbb{E}[\boldsymbol{\Delta}_l^{K=\infty}] = \boldsymbol{g}_l^{K=\infty}, \tag{83}$$

where $\boldsymbol{g}_l^{K=\infty}$ is defined in Eq. (18).

By Eqs. (81) and (83), we thus have

$$\mathbb{E}\left\|\boldsymbol{\Delta}_t^{K=\infty}\right\|^2 = C_t \cdot \mathbb{E}\left\|\boldsymbol{\Delta}_{t-1}^{K=\infty}\right\|^2 + D_t, \tag{84}$$

where $C_t$ denotes the coefficient of $\left\|\boldsymbol{\Delta}_{t-1}^{K=\infty}\right\|^2$ and $D_t$ denotes the remaining parts. The specific expressions of $C_t$ and $D_t$ are in Eqs. (19) and (20). Applying Eq. (84) recursively, we thus have Eq. (21).

**Underparameterized situation**

In the underparameterized situation, the convergence point of local steps in each round corresponds to the solution that minimizes the training loss, i.e.,

$$\begin{aligned}
\hat{\boldsymbol{w}}_{(i),t}^{K=\infty} &= (\mathbf{X}_{(i),t}\mathbf{X}_{(i),t}^\top)^{-1}\mathbf{X}_{(i),t}\boldsymbol{y}_{(i),t} \\
&= (\mathbf{X}_{(i),t}\mathbf{X}_{(i),t}^\top)^{-1}\mathbf{X}_{(i),t}(\mathbf{X}_{(i),t}^\top\boldsymbol{w}_{(i),t} + \boldsymbol{\epsilon}_{(i),t}) \text{ (by Eq. (2))} \\
&= \boldsymbol{w}_{(i),t} + (\mathbf{X}_{(i),t}\mathbf{X}_{(i),t}^\top)^{-1}\mathbf{X}_{(i),t}\boldsymbol{\epsilon}_{(i),t}.
\end{aligned}$$

Also recalling Eqs. (3) and (7), we thus have

$$\boldsymbol{\Delta}_t^{K=\infty} = \frac{1}{\sum_{i\in[m]} n_{(i),t}} \sum_{i\in[m]} n_{(i),t}(\boldsymbol{\gamma}_{(i),t} - (\mathbf{X}_{(i),t}\mathbf{X}_{(i),t}^\top)^{-1}\mathbf{X}_{(i),t}\boldsymbol{\epsilon}_{(i),t}). \tag{85}$$

For any $i,j \in [m]$, because $\boldsymbol{\epsilon}_{(j),t}$ is independent of $\mathbf{X}_{(i),t}$ and $\boldsymbol{\epsilon}_{(i),t}$, and also because $\boldsymbol{\epsilon}_{(j),t}$ has zero mean (by Assumption 1), we have

$$\mathbb{E}\left[\boldsymbol{\gamma}_{(j),t}^\top(\mathbf{X}_{(i),t}\mathbf{X}_{(i),t}^\top)^{-1}\mathbf{X}_{(i),t}\boldsymbol{\epsilon}_{(i),t}\right] = 0 \text{ for all } i,j \in [m],$$

$$\mathbb{E}\left[\left(\mathbf{X}_{(j),t}\mathbf{X}_{(j),t}^\top)^{-1}\mathbf{X}_{(j),t}\boldsymbol{\epsilon}_{(j),t}\right)^\top (\mathbf{X}_{(i),t}\mathbf{X}_{(i),t}^\top)^{-1}\mathbf{X}_{(i),t}\boldsymbol{\epsilon}_{(i),t}\right] = 0 \text{ for all } i \neq j.$$

Thus, by Eq. (85), we have

$$\begin{aligned}
\mathbb{E}\left\|\boldsymbol{\Delta}_t^{K=\infty}\right\|^2 &= \frac{1}{(\sum_{i\in[m]} n_{(i),t})^2} \sum_{i\in[m]} n_{(i),t}^2 \left(\left\|\boldsymbol{\gamma}_{(i),t}\right\|^2 + \mathbb{E}\left\|(\mathbf{X}_{(i),t}\mathbf{X}_{(i),t}^\top)^{-1}\mathbf{X}_{(i),t}\boldsymbol{\epsilon}_{(i),t}\right\|^2\right) \\
&\quad + \frac{1}{(\sum_{i\in[m]} n_{(i),t})^2} \sum_{i\in[m]}\sum_{j\in[m]\setminus\{i\}} n_{(i),t}n_{(j),t}\boldsymbol{\gamma}_{(i),t}^\top\boldsymbol{\gamma}_{(j),t} \\
&= \left\|\frac{\sum_{i\in[m]} n_{(i),t}\boldsymbol{\gamma}_{(i),t}}{\sum_{i\in[m]} n_{(i),t}}\right\|^2 + \frac{\sum_{i\in[m]}\frac{n_{(i),t}^2 p\sigma_{(i),t}^2}{n_{(i),t}-p-1}}{(\sum_{i\in[m]} n_{(i),t})^2} \text{ (by Eq. (28) in Lemma 3).}
\end{aligned}$$

We thus have proven Eq. (22).

The result of this theorem thus follows. $\qquad\square$

## F  A TABLE FOR NOTATIONS

We provide a table of some important notations used in this paper.

## G  MORE RELATED WORK

**Federated Learning.** Federated Learning (FL) has emerged as a pivotal distributed learning framework, harnessing the collaborative power of multiple clients to learn a shared model (Li et al., 2019;

| symbol | meaning |
|---|---|
| $n_{(i),t}$ | number of training samples |
| $\tilde{n}_{(i),t}$ | batch size |
| $p$ | number of parameters |
| $\sigma_{(i),t}$ | noise level |
| $\mathbf{X}_{(i),t}$ | matrix for input of training samples |
| $\boldsymbol{y}_{(i),t}$ | vector for output of training samples |
| $\boldsymbol{\epsilon}_{(i),t}$ | vector for noise of training samples |
| $\hat{\boldsymbol{w}}_0$ | the pre-trained parameters (initialization) |
| $\boldsymbol{w}^*$ | the learning target |
| $\boldsymbol{w}_{(i),t}$ | the ground-truth of agent $i$ at round $t$ |
| $\hat{\boldsymbol{w}}_{(i),t}^{K=1}, \hat{\boldsymbol{w}}_{(i),t}, \hat{\boldsymbol{w}}_{(i),t}^{K=\infty}$ | the local learning result of agent $i$ at round $t$ |
| $\hat{\boldsymbol{w}}_{(i),t,k}$ | learning result after $k$-th batch (for $K < \infty$ case) |
| $\hat{\boldsymbol{w}}_{\text{avg},t}^{K=1}, \hat{\boldsymbol{w}}_{\text{avg},t}^{K<\infty}, \hat{\boldsymbol{w}}_{\text{avg},t}^{K=\infty}$ | the FedAvg result at round $t$ |
| $\left\|\boldsymbol{\Delta}_t^{K=1}\right\|^2, \left\|\boldsymbol{\Delta}_t^{K<\infty}\right\|^2, \left\|\boldsymbol{\Delta}_t^{K=\infty}\right\|^2$ | model error |
| $\left\|\boldsymbol{\Delta}_0\right\|^2$ | initial (pre-trained) model error |
| $\alpha_{(i),t}$ | learning rate (step size) |
| $\boldsymbol{\gamma}_{(i),t}$ | measurement of heterogeneity |

Table 2: Table for some notations.

Yang et al., 2019a; Kairouz et al., 2019). Since its inception, FL systems have demonstrated increasing prowess, effectively handling diverse forms of heterogeneity in data, network environments, and worker computing capabilities. A multitude of prevalent FL algorithms, including FedAvg (McMahan et al., 2016) and its various adaptations (Li et al., 2018; Zhang et al., 2020; Karimireddy et al., 2020b;a; Acar et al., 2021; Yang et al., 2021; 2022), have contributed to the advancement of this framework. However, it is worth pointing out that these works only provide insights on the convergence in optimization while lacking the exploration of generalization performance for FL.

**Generalization performance of FL.** In the literature, there has been relatively limited studies on the generalization of FL. We categorize these works into three distinct classes. The first line of works employs the traditional analytical tools from statistical learning. Yuan et al. (2022) assumes that clients' data distributions are drawn from a meta-population distribution. Accordingly, they define two generalization gaps in FL: one is the participation generalization gap to measure the difference between the empirical and expected risk for participating clients, the same as the definition in classic statistical learning; the second is the non-participation generalization gap, which measures the difference of the expected risk between participating and non-participating clients. Following this two-level distribution framework, sharper bounds are provided (Hu et al., 2023). Zhao et al. (2023) utilized the Probably Approximately Correct (PAC) Bayesian framework to investigate a tailored generalization bound for heterogeneous data in FL. More recently, Sun et al. (2023) studied FL generalization by data heterogeneity through algorithmic stability and Sefidgaran et al. (2023) established PAC-Bayes and rate-distortion theoretic bounds on the generalization error. More works utilize similar tools to study the generalization in FL (Chor et al., 2023; Barnes et al., 2022; Sefidgaran et al., 2022; Huang et al., 2021). The second class of works studied the training dynamic near a manifold of minima and the effect of stochastic gradient noise on generalization. They used "sharpness" as a useful tool for generalization. Caldarola et al. (2022) and Shi et al. (2023) investigated the generalization behavior through the lens of the geometry of the loss and Hessian eigenspectrum, linking the model's lack of generalization capacity to the sharpness of the solution under ideal client participation. Based on the sharpness, Qu et al. (2022) proposed a momentum algorithm with better generalization. Gu et al. (2022) utilizes the stochastic differential equation (SDE) approximation to study the long-term behavior of the learning process. They showed that utilizing local steps always exhibits better generalization under appropriate conditions, including a sufficiently small learning rate, enough number of communication rounds, and the local steps being tuned. All of these existing studies primarily yield asymptotic results by focusing on domain changes or describing limiting behavior such as sufficiently large communication rounds and fine-tuned local steps. Consequently, they do not establish a direct, quantifiable relationship

that demonstrates how key factors—namely, data heterogeneity, the local update process, and the communication round—affect the generalization performance of FL.

**Model Averaging.** Model averaging, as discussed in works such as (DOBRIBAN & SHENG, 2021; Kamp et al., 2019; 2014), shares a resemblance to federated learning due to the commonality of employing a periodic averaging process. However, a fundamental distinction lies in the problem setting: federated learning assumes different local data distributions for each client, while model averaging assumes that the data in each client is sampled from one identical distribution. In this context, it's noteworthy that our results can be regarded as a degeneration to their setting under the assumption of independent and identically distributed (IID) data.

