# OpenReview forum: "Understanding the Theoretical Generalization Performance of Federated Learning"
_ICLR.cc/2024/Conference — Submitted to ICLR 2024_

### Official Review · Reviewer_snNy · 2023-10-30

**Soundness:** 2 fair
**Presentation:** 1 poor
**Contribution:** 1 poor
**Rating:** 3
**Confidence:** 4

**Summary:**

The paper analyzes the convergence of models in federated learning to the optimum in a convex scenario with standard normal features and Gaussian noise.

**Strengths:**

- generalization bounds for Federated Learning are an interesting and largely open problem (the only existing generalization bound for FL for deep learning I know relies on the NTK framework [6]).
- the paper analyzes both stationary and non-stationary targets

**Weaknesses:**

- the contribution beyond existing results is unclear
- relevant related work is not discussed
- the results are presented in a convoluted and unintelligible way with an unnecessary function $\mathcal{F}$ that hides the actual result and is not interpretable to me
- there are no experiments (e.g., on simulated data) to evaluate the tightness of the bounds
- the proofs provided in the Appendix are not well presented

**Questions:**

**Questions:**

- How do the results for $K>1$ in this paper relate to the results on model averaging for linear regression in [5]?
- Why do you need the assumption of standard normal features? While Gaussian noise is a common assumption in the analysis of linear regression, this assumption appears to strong to me. Even the analysis in [1] only assumes that data is generated by a process using a covariance operator and a random variable whose components are independent and subgaussian. Shouldn't such an already strong assumption suffice?
- Why do you not use (a variant of) the notion of (shifting-)regret to analyse the non-stationary case? For this case (i.e., online learning), the (shifting-)regret is typically used as a success measure, since it captures the nature of the task better than the loss [3].

**Detailed Comments:**

- the paper uses a myriad of newly defined symbols which makes it very hard to follow the writing, if one hasn't learned the symbols by heart. The presentation could be greatly improved by reminding the reader what certain symbols stand for.
- The K=1 case of FL is equivalent to simple distributed SGD, i.e., it can be considered to be a centralized SGD with larger batch size and smaller learning rate  [cf. Prop. 2 in 8]. For convex problems, this has been extensively studied [2]. How does the case in this paper differ?
- Since the paper investigates convex problems, standard convergence results even for non-convex FL (where, e.g., it is shown that $||\nabla L|| = 0$) imply that $w^* = w$. This includes, e.g., [4, 12, 14]. Please elaborate on the contribution beyond these works.
- The problem of heterogeneous local data in FL and its impact on convergence has also been extensively studied [7, 11, 15]. Please elaborate on the contribution beyond these works, in particular to Lemma 3 in [6] which seems to be a generalization of the results in this paper for deep learning.
- The results for the overparameterized case are not discussed wrt. benign overfitting for linear models [1]. The results of [1] should at least be discussed for the K=1 case, where FL boils down to distributed SGD.
- model averaging for stationary and non-stationary models in convex environments have been studied extensively [9, 10, 13]. How do the results in this paper relate to this previous work?



[1] Bartlett, Peter L., et al. "Benign overfitting in linear regression." Proceedings of the National Academy of Sciences 117.48 (2020): 30063-30070.\
[2] Boyd, Stephen P., and Lieven Vandenberghe. Convex optimization. Cambridge university press, 2004.\
[3] Cesa-Bianchi, Nicolo, and Gábor Lugosi. Prediction, learning, and games. Cambridge university press, 2006.\
[4] Charles, Zachary, and Jakub Konečný. "Convergence and accuracy trade-offs in federated learning and meta-learning." International Conference on Artificial Intelligence and Statistics. PMLR, 2021.\
[5] Edgar Dobriban. Yue Sheng. "Distributed linear regression by averaging." Ann. Statist. 49 (2) 918 - 943, April 2021. https://doi.org/10.1214/20-AOS1984\
[6] Huang, Baihe, et al. "Fl-ntk: A neural tangent kernel-based framework for federated learning analysis." International Conference on Machine Learning. PMLR, 2021.\
[7] Li, Xiang, et al. "On the convergence of fedavg on non-iid data." arXiv preprint arXiv:1907.02189 (2019).\
[8] Kamp, Michael, et al. "Efficient decentralized deep learning by dynamic model averaging." Machine Learning and Knowledge Discovery in Databases: European Conference, ECML PKDD 2018.\
[9] Kamp, Michael. Black-box parallelization for machine learning. Diss. Universitäts-und Landesbibliothek Bonn, 2019.\
[10] Kamp, Michael, et al. "Communication-efficient distributed online prediction by dynamic model synchronization." Machine Learning and Knowledge Discovery in Databases: European Conference, ECML PKDD 2014.\
[11] Karimireddy, Sai Praneeth, et al. "Scaffold: Stochastic controlled averaging for federated learning." International conference on machine learning. PMLR, 2020.\
[12] Koloskova, Anastasiia, Sebastian U. Stich, and Martin Jaggi. "Sharper convergence guarantees for asynchronous sgd for distributed and federated learning." Advances in Neural Information Processing Systems 35 (2022): 17202-17215. \
[13] Mcdonald, Ryan, et al. "Efficient large-scale distributed training of conditional maximum entropy models." Advances in neural information processing systems 22 (2009).\
[14] Yu, Hao, Sen Yang, and Shenghuo Zhu. "Parallel restarted SGD with faster convergence and less communication: Demystifying why model averaging works for deep learning." Proceedings of the AAAI Conference on Artificial Intelligence. Vol. 33. No. 01. 2019.\
[15] Yuan, Xiaotong, and Ping Li. "On convergence of FedProx: Local dissimilarity invariant bounds, non-smoothness and beyond." Advances in Neural Information Processing Systems 35 (2022): 10752-10765.

**Details Of Ethics Concerns:**

no ethics concerns.

---

> ### Author Response · Authors · 2023-11-22
>
> Thank you for your review. We have addressed the concerns in the following clarification and incorporated related discussions into the paper, with the main changes highlighted in blue text.
> >the contribution beyond existing results is unclear; relevant related work is not discussed
>
> General Response about contribution: We appreciate the author’s comments on numerous existing papers in online learning, model averaging, and classic convex optimization. However, it is essential to highlight that federated learning constitutes an independent subarea due to its intricate interplay between heterogeneous data and local update steps throughout the learning process. The distinctive settings of federated learning, characterized by the diversity of data sources and the incorporation of local update steps, differ significantly from other contexts. Our objective is to explicitly quantify the influence of data heterogeneity, local update steps, and the total number of communication rounds on the generalization performance of FL. This area remains under-explored in federated learning. More specifically:
>
> 1). The unique setting of federated learning in deep learning (over-parameterization) is significantly different compared to works in online learning, model averaging, and classic convex optimization [2, 5, 8, 9, 10, 13].
>
> 2). While many works study the convergence of federated learning, including those mentioned by the reviewer [4, 7, 11, 12, 14, 15], our focus is on *generalization*.
>
> 3). The only paper mentioned by the reviewer that includes generalization analysis is [6]. This generalization extends from centralized learning using the Probably Approximately Correct (PAC) framework, which falls into the first class of related works in our paper. Please see Page 1 in our paper (“From a theoretical perspective, there has been relatively limited studies in addressing this question. We can categorize existing explorations into two classes. The first line of work employs the traditional analytical tools from statistical learning, such as the “probably approximately correct” (PAC) framework. These works focus on the domain changes due to the data and system heterogeneity…”). However, it's important to note that these works, including [6], do not provide an explicit relationship showing how critical factors in FL, such as the local update process, the number of communication rounds, and data heterogeneity, collectively affect FL's generalization. In other words, existing works have not fully explored the influence of these factors on the generalization performance of FL, which is the primary goal of our paper.
>
> Related works: We appreciate the reviewer’s comment for various previous works studying the convergence of federated learning, and we will include them in the revision for a self-contained introduction of federated learning. It is essential to note, however, that our primary emphasis lies on the generalization of federated learning, for which we have meticulously presented an in-depth exploration of related works in our submission.
> >there are no experiments (e.g., on simulated data) to evaluate the tightness of the bounds
>
> Response: Our Figs. 1, 2, and 3 are the numerical results for three cases to verify our three theorems. As shown in these figures, each marker is the average of 20 simulation runs and the curves are theoretical values from corresponding theorems. All these experiments show the tightness of our theoretical results.
> >the results are presented in a convoluted and unintelligible way with an unnecessary function $\mathcal{F}$ that hides the actual result and is not interpretable to me
>
> Response: We explain the meaning of  $\mathcal{F}$ in Section 2.5. In short,  $\mathcal{F}$ corresponds to the general-term formula for a linear recurrence relation. Notice that we consider FL in multiple rounds, which is essentially a recurrence relation. Meanwhile, to help the readers understand our results, we provide expressions of the results in the simple case that does not contain $\mathcal{F}$.
> >the proofs provided in the Appendix are not well presented
>
> Response: Thanks for your comments. However, it is unclear to us which parts of the proofs in the Appendix the reviewer referred to are not well presented. We would highly appreciate it if the reviewer can pinpoint to a certain proof, and we would be more than happy to make improvements.

---

> > ### Author Response · Authors · 2023-11-22
> >
> > >How do the results for $K>1$ in this paper relate to the results on model averaging for linear regression in [5]?
> >
> > Response: Thanks for your question. Although both works study the linear regression model, there exist two major differences between our paper and [5]: 1) we study the over-parameterized setting (p > n) while [5] studies the under-parameterized case (see Page 5 in [5]). 2) we study the generalization of iterative processes of multiple model averages (i.e., the federated learning process) while [5] studies exactly one-step weighted averaging (see Abstract and Sec 3 in [5]).
> > >Why do you need the assumption of standard normal features? While Gaussian noise is a common assumption in the analysis of linear regression, this assumption appears too strong to me. Even the analysis in [1] only assumes that data is generated by a process using a covariance operator and a random variable whose components are independent and subgaussian. Shouldn't such an already strong assumption suffice?
> >
> > Response: Thanks for your question. We use the assumption of standard normal features to make our analysis and results more concise and tractable. Specifically, with standard normal features, our theories give exact values of the generalization performance, instead of upper or lower bounds (which is less precise than the exact value). Compared with [1] where the authors used sub-Gaussian instead of standard Gaussian distributions, the results in [1] are only the upper bounds of the generalization error.
> > >Why do you not use (a variant of) the notion of (shifting-)regret to analyse the non-stationary case? For this case (i.e., online learning), the (shifting-)regret is typically used as a success measure, since it captures the nature of the task better than the loss [3].
> >
> > Response: Our work proposes a unified framework to analyze the generalization of general federated learning, which includes the case of non-stationary data for each client. However, the goal for federated learning is to find a single solution given all the data from all clients. That is, the goal of federated learning is to find a solution that can approximate the global solution x*. Thus, the difference between the solution x_t and x* is used as the metric for generalization. This is different from online learning, in which regret is used to quantify how well an algorithm adapts to changes in the environment over time. Meanwhile, the regret can be easily calculated based on our current results by summing up the expressions over $t$.
> > >the paper uses a myriad of newly defined symbols which makes it very hard to follow the writing, if one hasn't learned the symbols by heart. The presentation could be greatly improved by reminding the reader what certain symbols stand for.
> >
> > Response: Thanks for your suggestion. In the revision, we added a table of notations in  Appendix F to help readers keep track of important notation.
> > >The K=1 case of FL is equivalent to simple distributed SGD, i.e., it can be considered to be a centralized SGD with larger batch size and smaller learning rate [cf. Prop. 2 in 8]. For convex problems, this has been extensively studied [2]. How does the case in this paper differ?
> >
> > Response: Since we considered heterogeneity and non-stationarity, the $K=1$ case of FL is not necessarily equivalent to a simple distributed SGD. Since we want to know how system parameters such as the number of rounds and heterogeneity affect the generalization performance, our analysis and results are much richer compared to and are not included by [2].
> > >Since the paper investigates convex problems, standard convergence results even for non-convex FL (where, e.g., it is shown that $||\nabla L|| = 0$) imply that $w^* = w$. This includes, e.g., [4, 12, 14]. Please elaborate on the contribution beyond these works.
> >
> > Response: Thanks for your comment. The major contribution in our work is that our focus is on the generalization of federated learning while these prior works only studied the convergence of federated learning.

---

> > > ### Author Response · Authors · 2023-11-22
> > >
> > > >The problem of heterogeneous local data in FL and its impact on convergence has also been extensively studied [7, 11, 15]. Please elaborate on the contribution beyond these works, in particular to Lemma 3 in [6] which seems to be a generalization of the results in this paper for deep learning.
> > >
> > > Response: Thanks for your comments. We note that our focus is the generalization of federated learning, while these papers study the convergence of federated learning. In [6], the authors studied the generalization of federated NTK (it seems that we couldn’t find Lemma 3 there, if we didn’t miss anything). The generalization of [6] is the extension of centralized learning, where it can be seen from the proof (Sec C in their appendix) that their results heavily rely on [R2]. From a high level, this work belongs to the first class of generalization as summarized in the introduction of our paper (Page 1 - 2). These works do not provide an explicit relationship to show how the critical factors in FL, (e.g., the local update process, the number of communication rounds, and data heterogeneity) affect the generalization of FL in general, which is the goal of our paper.
> > >
> > > [R2] Arora, Sanjeev, et al. "Fine-grained analysis of optimization and generalization for overparameterized two-layer neural networks." International Conference on Machine Learning. PMLR, 2019.
> > > >The results for the overparameterized case are not discussed wrt. benign overfitting for linear models [1]. The results of [1] should at least be discussed for the K=1 case, where FL boils down to distributed SGD.
> > >
> > > Response: Thanks for your comments. However, we do have comparisons with the benign overfitting of classical (single-task) linear models in Eq. (25) for $K=\infty$ case, which leads to the insight 4 in the paper (FL can alleviate the null risk compared to the classical single-task/agent benign overfitting). Notice that $K=1$ is the case of one-step local gradient update in each round, which does not overfit the training data set (and thus does not belong to benign overfitting).
> > >
> > > >model averaging for stationary and non-stationary models in convex environments have been studied extensively [9, 10, 13]. How do the results in this paper relate to this previous work?
> > >
> > > Response: Thanks for your questions. Model averaging [9,10,13] shares some similarities with federated learning for the same periodic averaging process. However, one fundamental difference in terms of problem settings is that federated learning typically has heterogeneous local data distributions at each client, while the data in each client for model averaging is assumed to be sampled from the same distribution (e.g., see Sec 2.1 in [10] and Sec 2.2 in [13]). Our results can relate and degenerate to the settings in [9,10,13] under the i.i.d. data setting.

---

> > ### Comment · Reviewer_snNy · 2023-11-22
> > **Reply to authors**
> >
> > Dear authors,
> >
> > In your reply 1) you claim that federated deep learning is significantly different to [8] and [9]. Both of them cover federated deep learning.
> >
> > In your reply 2) you claim that your focus is on generalization, whereas the cited papers focus on convergence. Generalization is defined as the difference between the risk and empirical risk. Your main results are about the distance to the optimal model in convex settings, though. As mentioned, such results on distance to the optimum are either directly or trivially covered by the mentioned works.

---

> > > ### Author Response · Authors · 2023-11-22
> > >
> > > Thanks for your prompt response. Our claim is that the unique setting of federated learning in deep learning (over-parameterization) is significantly different from [8,9]. We agree with the reviewer that federated learning shares the similar period model averaging with previous works [8,9]. However, it is worth pointing out the main differences in the setting: 1) We study the over-parameterization setting tailored to deep learning. The setting of previous works, including [8,9], simply used an abstract function f without showing any specific features for deep learning. 2) One of our focuses is the impact of diverse data sources among clients. That is, each client has local data sampled from different distributions. This is different from [8] that assumes i.i.d. data are drawn from the same time-varying distribution (Sec 2 in [8]). 3) Moreover, we study **generalization** (test loss) rather than optimization. In other words, our target is to minimize the expected risk $E_{\xi \sim P} [f(x, \xi)]$, as opposed to the goal of optimization to minimize the empirical risk $\sum_{i \in [n]} f(x, \xi_i)$ with n data samples. These three differences differ significantly from other contexts.
> > >
> > > In our paper, what we meant by the generalization performance is actually the test performance. We note that the test error is the same as generalization error when the training error is zero, which is the case of overfitting on over-parameterization deep learning. We also added Lemma 6 in the revision to clarify our performance metric. We want to emphasize that the optimal solution that minimizes the empirical loss (global optimal of training) is not always the true “optimal” solution (i.e., ground truth $w^*$), especially in the context of overfitting. In fact, the whole line of research on overfitting and benign overfitting is motivated by this critical difference. This is exactly why we want to study the test performance, rather than the convergence to the global optimal solution of training.

---

### Official Review · Reviewer_ocpm · 2023-10-31

**Soundness:** 2 fair
**Presentation:** 2 fair
**Contribution:** 3 good
**Rating:** 5
**Confidence:** 3

**Summary:**

The paper studies the generalization error of the Federated Learning algorithm under a simple linear model and i.i.d. data. The results characterize the behavior of the generalization error in FL with respect to many factors, including the heterogeneity of the data, the number of local updates and communication, and by considering the model size, in under-parametrized and over-parametrized regime.

**Strengths:**

Very little is known about the theoretical understanding of generalization performance in FL. This paper is one of the first to address this issue. The authors provide several interesting insights into the generalizability of FL and study the effects of various parameters, such as the heterogeneity of the data, the number of local updates and communication, etc. Moreover, the authors provided the first explanation of the double-descent behavior in FL setup. The technical level of the paper also looks high, although I have not verified it carefully.

**Weaknesses:**

Despite its clear strengths, the paper suffers from several problems, which are listed below. I would increase my rating if the major concerns can be addressed, as the paper is among the few works that provides insight into the generalization performance of FL.

1.	The model, i.e. the consideration of a linear model with i.i.d. Gaussian data, is very simple and unrealistic. However, as mentioned by the authors, this is now a commonly used model as a first step in understanding the theory.

2.	In particular, there is an inherent alignment between empirical risk and generalization error in the considered setup. That is, they are somehow simultaneously minimized, which is in sharp contrast to most real-world scenarios. In fact, throughout the paper, the authors study "model error," a quantity that one would also study for empirical risk. For this reason, I am not sure if the findings on generalization behavior using this setup can be extended or useful for realistic setups.


3.	In addition, the authors stated that the model error can be shown to be equal to the expected test error for noise-free data. What happens if the data is not noise free? (I guess this is the interesting case, right?). How does the "model error" relate to the generalization performance in this case?

4.	Continuing the point 2, a good “generalization bound” typically decreases with the total number of used symbol (here, it would be $m\times n \times t$ at the end of iteration $t$, in the simple case). How does the provided bounds behave with these quantities?


5.	It was observed and partially shown in previous papers (e.g. Gu et al. 2022 or arXiv:2306.05862) that the generalization error of the federated learning is smaller than that of centralized one (i.e. if we keep $m\times n$ fixed). Do authors observe similar phenomena using these theoretic results?

6.	All plots provide the numerical evaluation of the bounds on the model errors. It would be useful to plot the estimated model error as well as the estimated generalization error in the plots to observe how well the provided bounds capture the correct behavior of both the model error and the generalization error.

7.	Some relevant references are missing, e.g. arXiv:2306.03824 and arXiv:2306.05862.

8.	The constants defined in equations (10) to (12) require some explanation and intuition. What is each term made of and what do they represent? In its current form, it's very hard to parse and understand. Similarly, theorem 1 is very hard to parse (except for the simple form of (14)). Similarly, for other theorems.

9.	Theorem 1 (and other results) suggest that increasing the learning rate $\alpha$ increases the model error. While this behavior makes sense from an empirical risk point of view, it's often the opposite for generalization behavior: larger learning rates, at least in some cases and for the classical centralized setup, lead to better generalization performance.

10.	Furthermore, Theorem 1 suggests that more heterogeneity leads to higher generalization error. This is also counterintuitive to me. After all, this is exactly what I would expect for empirical risk, but I would expect the opposite for generalization.

11.	If I'm not mistaken, equation (16) (and Proposition 1) is developed for the "constant learning rate" scenario. While for this scenario the existence of an optimal $K$ makes sense, I was wondering what would be the behavior with respect to $K$ for the carefully tuned decreasing learning rates?

12.	The bound of Theorem 3 interestingly suggests a double descent phenomenon. However, the paper would greatly benefit from a numerical verification of this.


13.	It is written that "In Figure 3, we plot the model error against p...". However, the caption of Figure 3 states that "The curves are theoretical values from Theorem 3". It is a bit confusing whether the exact "model error" is plotted or the established bound on it? It would be useful to plot both (as well as the generalization error).

14.	The proofs are very long and technical, and it takes a lot of time to go through them. I think it would be helpful (and perhaps even necessary) to give a proof sketch in the main text, explaining the main steps, ideas, and tools used from the literature, and what the authors have added. In the current format, it is not clear what technical novelty the authors bring to this work.

15.	The supplementary material (in particular Appendix A) may be more appropriately organized, with a table of contents to guide the reader to the various sections.

16.	It would be good to summarize the provided insights in the contribution part of the introduction.

**Questions:**

Mentioned above.

---

> ### Author Response · Authors · 2023-11-22
>
> Thank you for your review. We have addressed the concerns in the following responses and incorporated related discussions into the paper, with the main changes highlighted in blue text.
> >1. The model, i.e. the consideration of a linear model with i.i.d. Gaussian data, is very simple and unrealistic. However, as mentioned by the authors, this is now a commonly used model as a first step in understanding the theory. 2. In particular, there is an inherent alignment between empirical risk and generalization error in the considered setup. That is, they are somehow simultaneously minimized, which is in sharp contrast to most real-world scenarios. In fact, throughout the paper, the authors study "model error," a quantity that one would also study for empirical risk. For this reason, I am not sure if the findings on generalization behavior using this setup can be extended or useful for realistic setups.
>
> Response: Thanks for your insightful comments regarding our model setting. For better readability, we also structure our response to your question accordingly as follows:
>
> **Simultaneously minimized risk and generalization erros:** We respectfully disagree that both empirical risk and the generalization error are simultaneously minimized. In the cases of $K=1$ and $K<\infty$, the empirical risk is not minimized because the SGD/GD update is stopped after K steps (i.e., early stop). In the case of $K=\infty$, the empirical risk is minimized but the generalization error is not, since Eq. (6) suggests a zero empirical risk but the generalization error shown by Theorem 3 is still positive.
>
> **“Model error”:** As we explained in the footnote of Page 4, the model error is equal to the expected test error for noise-free test data. This quantity is widely used to indicate the generalization performance in a linear model setup.
>
> >3. In addition, the authors stated that the model error can be shown to be equal to the expected test error for noise-free data. What happens if the data is not noise free? (I guess this is the interesting case, right?). How does the "model error" relate to the generalization performance in this case?
>
> Response: Thanks for your question. The difference in the expected test error between the case of noise-free test data and the case of noisy test data is only the noise level (a constant that is irrelevant to the learning process). In the revision, we added Lemma 6 to make this claim rigorous.
>
> >4. Continuing the point 2, a good “generalization bound” typically decreases with the total number of used symbols (here, it would be $m\times n \times t$ at the end of iteration $t$, in the simple case). How does the provided bounds behave with these quantities?
>
> Response: Thanks for your question. It appear sthat there are some misunderstandings in here. First, we want to clarify that our characterization of the generalization error is not a bound. Rather, it is the **exact value** of the expected generalization error. The correctness of our characterization is also validated by Figs. 1-3 that analytical values perfectly match simulation values. Second, the expressions in Theorems 1~3 contain $m$, $n$, and $t$ explicitly. For example, in Eq. (14), both H and G have $mn$ in their denominator and thus decrease with respect to $mn$. When the learning rate is small, we have $H<1$, which suggests the first term of Eq. (14) also decreases with $t$. The second term of Eq. (14) (noise term) increases with $t$ due to the accumulation of the noise effect over time.
> >5. It was observed and partially shown in previous papers (e.g. Gu et al. 2022 or arXiv:2306.05862) that the generalization error of the federated learning is smaller than that of centralized one (i.e. if we keep $m\times n$ fixed). Do authors observe similar phenomena using these theoretic results?
>
> Response: Thanks for your question. Yes, we do observe similar phenomena in the case of $K=\infty$. As mentioned in our Insight 4, compared to the centralized one, the “null risk” can be alleviated and can achieve a smaller generalization error.

---

> > ### Author Response · Authors · 2023-11-22
> >
> > >6. All plots provide the numerical evaluation of the bounds on the model errors. It would be useful to plot the estimated model error as well as the estimated generalization error in the plots to observe how well the provided bounds capture the correct behavior of both the model error and the generalization error.
> >
> > Response: Thanks for the suggestions. Note that we have already done so in the current figures. In Figs 1~3, the points (markers) are the numerical evaluations, and the curves (without the markers) are the analytical values. As these markers are very near to the curves (the curves almost pass through points exactly), the correctness of our analytical results is indeed verified.
> >
> > > 7. Some relevant references are missing, e.g. arXiv:2306.03824 and arXiv:2306.05862.
> >
> > Response: We appreciate the reviewer’s comment about the references, and we will include them in the revision. These two works use traditional analytical tools from statistical learning, such as stability, PAC-Bayes and rate-distortion, to establish the bounds. However, we note that these traditional tools *cannot* explicitly quantify the influence of data heterogeneity, local update steps, and the total number of communication rounds on the generalization performance of FL, which is part of the motivation of our work.
> > >8. The constants defined in equations (10) to (12) require some explanation and intuition. What is each term made of and what do they represent? In its current form, it's very hard to parse and understand. Similarly, theorem 1 is very hard to parse (except for the simple form of (14)). Similarly, for other theorems.
> >
> > Response: Thanks for your questions. For all theorems, we have provided their simplified versions to make them easy to interpret. For example, $H$ and $G$ in the simplified form Eq. (14) corresponds to $H_t$ and $G_t$ in Eqs. (10)(11), respectively. The additional terms in Eqs (10)(11) are due to the non-stationarity (that is why we have the subscript $t$ in $H_t$ and $G_t$) and the asymmetric heterogeneity ($\sum_j \gamma_{(j),t}\neq 0$, so there will be more terms about $\gamma_{(j),t}$ that won’t be canceled out).
> > >9. Theorem 1 (and other results) suggest that increasing the learning rate $\alpha$ increases the model error. While this behavior makes sense from an empirical risk point of view, it's often the opposite for generalization behavior: larger learning rates, at least in some cases and for the classical centralized setup, lead to better generalization performance.
> >
> > Response: We respectfully disagree that Theorem 1 suggests increasing the learning rate $\alpha$ always increases the model error (i.e., generalization error). In Eq. (14), a larger $\alpha$-value leads to a larger $G$ (the term for noise and heterogeneity). However, $H$ is a quadratic function with respect to $\alpha$. When there is no noise and no heterogeneity (i.e., $G=0$), the optimal $\alpha$ that can minimize the generalization error (i.e., minimize $H$) should equal to $\frac{mn}{mn+p+1}$. In this situation, when the learning rate increases from 0 to $\frac{mn}{mn+p+1}$, the generalization performance becomes smaller. Thus, there is no contradiction between our theoretical results and empirical observations.
> > >10. Furthermore, Theorem 1 suggests that more heterogeneity leads to higher generalization error. This is also counterintuitive to me. After all, this is exactly what I would expect for empirical risk, but I would expect the opposite for generalization.
> >
> > Response: Thanks for your comments. In general, heterogeneity has been observed to be a negative factor for generalization. For example, the paper mentioned by the reviewer in Comment 7 (arXiv:2306.03824) shows the same conclusion that heterogeneity leads to higher generalization error (Corollary 1 in arXiv:2306.03824).
> > >11. If I'm not mistaken, equation (16) (and Proposition 1) is developed for the "constant learning rate" scenario. While for this scenario the existence of an optimal $K$ makes sense, I was wondering what would be the behavior with respect to $K$ for the carefully tuned decreasing learning rates?
> >
> > Response: Thanks for your question. We agree that it is an interesting topic to further investigate whether an optimal $K$ exists when the learning rates are carefully tuned for different $K$. Meanwhile, we also note that the use of constant learning rates is the most common practice in training. Since our expression of the generalization performance contains $K$ and learning rate explicitly, we can replace $\alpha$ by a function of $K$ and then analyze the monotonicity of the whole expression. Depending on how $\alpha$ changes with $K$, the conclusion might be different. Due to the goal of the current paper is to take a first attempt to understand the generalization error of federated learning, we leave this to future work.

---

> > > ### Author Response · Authors · 2023-11-22
> > >
> > > >12. The bound of Theorem 3 interestingly suggests a double descent phenomenon. However, the paper would greatly benefit from a numerical verification of this.
> > > 13. It is written that "In Figure 3, we plot the model error against p...". However, the caption of Figure 3 states that "The curves are theoretical values from Theorem 3". It is a bit confusing whether the exact "model error" is plotted or the established bound on it? It would be useful to plot both (as well as the generalization error).
> > >
> > > Response: Thanks for your suggestions and questions. In Figs 1~3, the markers/points are simulated values while the curves are analytical values. In other words, the curves are not drawn by connecting the markers/points but are directly calculated by the analytical expressions.
> > > >14. The proofs are very long and technical, and it takes a lot of time to go through them. I think it would be helpful (and perhaps even necessary) to give a proof sketch in the main text, explaining the main steps, ideas, and tools used from the literature, and what the authors have added. In the current format, it is not clear what technical novelty the authors bring to this work.
> > >
> > > Response: Thanks for your suggestion. Due to the page limit, it is difficult to provide proof sketches for all three main theorems in the main body. The main technical tools we used are listed in Appendix A. Specifically, Lemmas 2~4 are from the literature, while Lemma 5 is the new tool we created. To help the readers understand the proof of Lemma 5, we also provide a geometric interpretation in Fig. 4.
> > > >15. The supplementary material (in particular Appendix A) may be more appropriately organized, with a table of contents to guide the reader to the various sections.
> > >
> > > Response: Thanks for your suggestion. In the revision, we have provided a table of contents to guide the reader to the various sections of the supplemental material. We have also added one paragraph at the beginning of Appendix A to make Appendix A more organized as follows:
> > > "Moreover, in Appendix A, we have provided some useful lemmas. Specifically, Lemma 1 is used to support the claim of the convergence speed in Insight 4. Lemmas 2 to 4 are some results about the Gaussian random matrices that can be found in the literature. We want to highlight that Lemma 5 is part of our technical novelty, which gives the exact values of terms related to the projection formed by each agent’s training inputs."
> > > >16. It would be good to summarize the provided insights in the contribution part of the introduction.
> > >
> > > Response: Thanks for your suggestion. We will add the following sentences in the introduction:
> > > Our results show interesting insights that 1) a good pre-trained model helps but only to some extent; 2) the effect of noise and heterogeneity accumulates but is still limited; 3) the optimal number of local updates sometimes exists; and 4) benign overfitting can exist in FL with alleviated null risk.

---

### Official Review · Reviewer_uk5o · 2023-11-01

**Soundness:** 2 fair
**Presentation:** 3 good
**Contribution:** 2 fair
**Rating:** 5
**Confidence:** 3

**Summary:**

This work investigates how data heterogeneity and the number of local update steps affects the performance of federated learning. The authors study the problem using a linear model with possibly time-varying ground truths. They quantify the $\ell_2$-estimation error in three different regimes: $K=1, K<\infty$, and $K=\infty$, where $K$ denotes the number of local update steps. The results characterize the optimal choice of $K$ in some cases and validate the “double descent” phenomena when $K = \infty$.

**Strengths:**

- This work considers a very general setting where the ground truth parameter of each agent may vary with time.
- This paper brings novel theoretical results of the effect of local update steps on federated learning. There is not much existing literature on this problem.
- The results provide insights on the optimal choice of local update steps for FL.
- The presentation is clear and well-organized, with plenty of discussion after each theoretical result.

**Weaknesses:**

- A major concern is the parameter defined by Equation (3). It does not seem like a suitable measurement for heterogeneity.
  Consider the case where $w_{(i), t} = w_{(j),t} = w^* + v$ with $\Vert v \Vert$ very large. In this case, all agents’ ground truth are the same, *i.e.* the data are homogeneous. However, the *level of heterogeneity* is very large by the definition in the paper.

**Questions:**

**Major**
Please see the Weaknesses section.

**Minor**
Is there a reference for Equation (25)?

---

> ### Author Response · Authors · 2023-11-22
>
> Thank you for your review. We have addressed the concerns in the following clarification and incorporated related discussions into the paper, with the main changes highlighted in blue text.
> >A major concern is the parameter defined by Equation (3). It does not seem like a suitable measurement for heterogeneity.
> Consider the case where $w_{(i), t} = w_{(j),t} = w^* + v$ with $\Vert v \Vert$ very large. In this case, all agents’ ground truth are the same, i.e. the data are homogeneous. However, the level of heterogeneity is very large by the definition in the paper.
>
> Response: Thanks for the comments. We note, however, that if every agent shares a very large and common $v$, then this quantity $v$ should be considered part of $w^*$ to accommodate such bias. Indeed, the choice of $w^*$ depends on all $w_{(i),t}$ (e.g., their average). In general, $w^*$ should be regarded as the “largest” overlapped part of $w_{(i),t}$. Therefore, our measurement for heterogeneity is still suitable.
>
> >Is there a reference for Equation (25)?
>
> Response: Thanks for your question. Yes, Eq. (25) can be found in [R1] (with changes of notations). Eq. (25) is also a special case of Eq. (23) in our Theorem 3 by letting $m=1$ and $t=1$.
>
> [R1] Mikhail Belkin, Daniel Hsu, and Ji Xu. Two models of double descent for weak features. SIAM Journal on Mathematics of Data Science, 2(4):1167–1180, 2020.

---

### Official Review · Reviewer_Rcvy · 2023-11-09

**Soundness:** 3 good
**Presentation:** 2 fair
**Contribution:** 1 poor
**Rating:** 3
**Confidence:** 4

**Summary:**

This paper studies optimizing a distributed linear regression problem with local SGD. The aim is to capture the effect of data heterogeneity and provide upper bounds on the $L_2$ estimation error of the ground truth predictor shared across the clients. To do so, the paper considers three settings: $K=1$ (i.e., MB-SGD update on the average objective), $K<\infty$ (i.e., vanilla local SGD), and $K=\infty$ (i.e., convergence on each machine between communication rounds). The first two settings are considered with a single pass on the data, while the third setting converges to the ERM solution on the machine. Closed-form upper bounds are provided in each of these settings for both
- **the online/non-stationary setting** (when the predictor on each machine changes across time); and
- **the stationary setting** (when the predictor on each machine is fixed over time but is potentially different across the machines).

Several insights are drawn from these closed-form expressions in a more straightforward setting, where the shared solution concept across the machines is the average of the machines' model for that communication round.

**Strengths:**

The paper is clearly written, making it easy to follow the mathematical details. All the expressions are fully worked out and the proofs are also easy to verify.

**Weaknesses:**

The paper claims to be general enough to provide results for the non-stationary setting. However, it is unclear if measuring the distance from $w^\star$ is useful in the non-stationary setting. In particular, it is never discussed what $w^\star$ is and why the machines would want to recover it in a general setting, as opposed to minimizing regret with respect to a fixed best model in hindsight. Due to this reason, it seems that the simple setting for presenting results, where $w^\star$ is the average optimum of different machines, is the only reasonable setting. As a result, the generality of the results in this paper is not very useful, without further motivation.

There are some issues even if we consider all the results in the simple setting. In the most challenging setting, that is, for a general $K<\infty$, $\bar{||\gamma||}^2$ is conveniently assumed to be zero. This implies that all the machines have the same optimum as $w^\star$. We already know the min-max optimal algorithms for quadratic functions in the homogeneous setting due to [Woodworth et al.](https://arxiv.org/abs/2002.07839). Thus, theorem 2 is not interesting. Similarly, Theorem 1 is not interesting as without any local update steps, there is no consensus error, and the update looks exactly like the mini-batch SGD update on the averaged objective across the machines. Resultwise, insights 1-3 are not interesting either. Finally, Theorem 3 is interesting but it is very easy to obtain using standard arithmetic.

Overall, I do not believe this paper is novel enough, and adds to the existing theory of local update methods.

**Questions:**

- How are the step sizes tuned in all the experiments? In particular, in Figure 2, are step sizes tuned separately for each experiment?

- (23) also has null risk, depending on the order of taking limits. For instance if $p\to \infty$ before $t\to\infty$, then there is a residual null risk. As a result, Insight 4 seems wrong. Figure 3 also seems to show this. What is the insight here then?

---

> ### Author Response · Authors · 2023-11-22
>
> We appreciate the reviewer's feedback. We would like to point out that the concerns are mostly due to misunderstanding and we have made corresponding modifications to the paper, which are highlighted in blue text in the revision. Our response is outlined below.
> >The paper claims to be general enough to provide results for the non-stationary setting. However, it is unclear if measuring the distance from w^* is useful in the non-stationary setting. In particular, it is never discussed what w^* is and why the machines would want to recover it in a general setting, as opposed to minimizing regret with respect to a fixed best model in hindsight. Due to this reason, it seems that the simple setting for presenting results, where w^* is the average optimum of different machines, is the only reasonable setting. As a result, the generality of the results in this paper is not very useful, without further motivation.
>
> Response: Thanks for your comments and suggestions. While letting $w^*$ be the average of different machines seems to be the most reasonable choice, there are some other possibilities in reality. For example, we may want to let $w^*$ to be the “smallest” (in certain criteria such as l2 norm) among different machines when the whole system’s performance depends on the machine with the smallest $w$.
>
> >There are some issues even if we consider all the results in the simple setting. In the most challenging setting, that is, for a general $K<\infty, \bar{||\gamma||}^2$ is conveniently assumed to be zero. This implies that all the machines have the same optimum as w^*. We already know the min-max optimal algorithms for quadratic functions in the homogeneous setting due to Woodworth et al.. Thus, theorem 2 is not interesting. Similarly, Theorem 1 is not interesting as without any local update steps, there is no consensus error, and the update looks exactly like the mini-batch SGD update on the averaged objective across the machines. Resultwise, insights 1-3 are not interesting either. Finally, Theorem 3 is interesting but it is very easy to obtain using standard arithmetic.
>
> Response: Thanks for your comments. It appears that there are some misunderstandings. First, our paper focuses on offering a theoretical understanding of the generalization performance of FL, not min-max optimal algorithms for quadratic functions. Second, Theorem 2 not only provides the results for the simple setting but also provides the general results with heterogeneity and non-stationarity. Third, we believe that results in Theorem 3 are non-trivial and the key part of the proof is Eqs. (78)~(80), which depends on our Lemma 5. The proof of Lemma 5 is complex and novel, which by no means can be obtained by standard arithmetic. We also use Figure 4 to give a geometric interpretation of the proof.
>
> >Overall, I do not believe this paper is novel enough, and adds to the existing theory of local update methods.
>
> Response: We would like to clarify and highlight that the main contribution and novelty of this paper is to derive exact expressions (instead of upper or lower bounds) of the generalization performance of FL for various settings of $K$, which provide meaningful insights. To the best of our knowledge, our work is the first of its kind in literature. That being said, we would highly appreciate it if the reviewer provides pointers to existing works that we are unaware of, and we would be happy to include and compare them in the revision of this paper.
>
> >Questions:
> How are the step sizes tuned in all the experiments? In particular, in Figure 2, are step sizes tuned separately for each experiment?
>
> Response: Thanks for the question. The (fixed) step sizes are set to 0.02 in all experiments.
>
> >(23) also has null risk, depending on the order of taking limits. For instance if $p\to\infty$ before $t\to\infty$, then there is a residual null risk. As a result, Insight 4 seems wrong. Figure 3 also seems to show this. What is the insight here then?
>
> Response: Thanks for your question. Again, it appears that there are some misunderstandings in how to interpret Insight 4. We note that Insight 4 only says the null risk is *alleviated* (not completely removed) by using more communication rounds and we explicitly discuss the issue of the speed w.r.t. $t$ and $p$ in the paragraph below Eq. (25). Hope this clarifies the confusion.

---

> > ### Comment · Reviewer_Rcvy · 2023-11-23
> >
> > I am not satisfied with the author's response and have decided to retain my score.
> >
> > Regarding the choice of $w^\star$ in the stationary but non-simple setting, the authors do not offer a convincing interpretation of $w^\star$ and why we should not care about a regret notion in the online setting. I am not sure why anything beyond the simple setting makes intuitive sense.
> >
> > I have checked again, and theorem 2 in the simple setting with $\gamma=0$ absolves to the homogenous setting, so the results are very incremental.
> >
> > Fixing the step size across experiments while varying the problem parameters or the number of machines makes little sense.
> >
> > Due to limited technical novelty and new insights, I do not believe the paper is fit for publication.

---

> > > ### Author Response · Authors · 2023-11-23
> > >
> > > > Regarding the choice of $w^*$ in the stationary but non-simple setting, the authors do not offer a convincing interpretation
> > >
> > > **Response**: We want to emphasize that $w^*$ is not something that can be chosen in FL. Instead, it is a description of the ground truth (i.e., the optimal value of the parameters). Depending on different problem setups, this optimal value can vary. For a theoretical work like ours, we don’t need to enumerate all possible/meaningful choices of the optimal value. Instead, we just give general theoretical results by leaving $w^*$ untouched in the expressions, so our results fit for any choices of $w^*$.
> > >
> > > ---
> > >
> > > > Why we should not care about a regret notion in the online setting
> > >
> > > **Response**: As we explained in the previous response, we have given the results of suboptimality for every round $t$ and the cumulative regret can be easily calculated by summing over $t$. In other words, our current metric is indeed finer than the cumulative regret.
> > >
> > > ---
> > >
> > > > I am not sure why anything beyond the simple setting makes intuitive sense. I have checked again, and theorem 2 in the simple setting with
> > >  absolves to the homogenous setting, so the results are very incremental.
> > >
> > > **Response**: We want to emphasize that our result is very general and the contribution is not limited by the simple case. For example, at the beginning of Theorem 2, we provide the general results for non-simple cases with heterogeneity and non-stationarity. The reason for providing the simple case results along with the general case is just to help readers have an intuitive understanding of our theorems. In other words, our results are not incremental because indeed we considered heterogeneity and non-stationarity.
> > >
> > > ---
> > >
> > > > Fixing the step size across experiments while varying the problem parameters or the number of machines makes little sense.
> > >
> > > **Response**: Our theoretical results do consider flexible step size. As a piece of evidence, the notation of the step size in Eqs. (10~12) is $\alpha_{(i),t}$ which can be different for different agents $i$ and rounds $t$. In our simulations, we use a fixed step size just for simplicity. However, our main contribution is on the theoretical side and the experiments/simulations are just used to validate/explain some interesting interpretations/insights derived from our theoretical analysis.
> > >
> > > ---
> > >
> > > > Due to limited technical novelty and new insights, I do not believe the paper is fit for publication.
> > >
> > > **Response**: To the best of our knowledge, our methods and insights on the generalization performance of FL are not seen in the existing literature. If the reviewer can provide specific literature that is similar to ours, we are also happy to compare it in the revision.

---

### Meta-Review · Area_Chair_tv7h · 2023-12-15

**Metareview:**

I did not read the paper myself, and as a meta-reviewer, I am not obliged to do so. I do so anyway in cases where it seems the reviews are of insufficient depth or quality, but this is not the case here. Hence, I am fully relying on the reviews, the authors-reviewers discussion, and the subsequent discussion among the reviewers.

The reviewers unanimously decided to reject the paper (scores 3, 3, 5, 5). They raised many specific critical comments, and the discussion phase did not resolve the issues in their eyes. Therefore, I have no choice but to recommend rejection.

Some of the issues raised:

- All reviewers agree that this paper offers little novelty. The results seem to be mostly trivial. It is unclear what new this work adds to the existing theory of local update methods.

- Some very closely relevant work is not discussed.

- The paper claims to be general enough to provide results for the non-stationary setting. However, it is unclear if measuring the distance from is useful in the non-stationary setting.

- The parameter defined in Equation (3) does not seem to be a good measure of heterogeneity.

- There are some issues even if we consider all the results in the simple setting. In the most challenging setting, that is, for a general $K<\infty$ case, the quantity $||\bar{\gamma}||^2$  is conveniently assumed to be zero. This implies that all the machines have the same optimum as $w^*$.

- Reviewer ocpm mentioned 16 specific weaknesses; the key ones remained not sufficiently addressed.

- Theorem 3 is interesting but it is very easy to obtain using standard arithmetic.

- The results are presented in a convoluted and unintelligible way

- There are no experiments (e.g., on simulated data) to evaluate the tightness of the bounds

- The proofs provided in the Appendix are not well presented

**Justification For Why Not Higher Score:**

Not a single reviewer wanted to accept the paper, despite a healthy discussion with the authors, and among reviewers. The reviews look reasonable. Hence, the paper can't be accepted.

**Justification For Why Not Lower Score:**

N/A

---

### Decision · Program_Chairs · 2024-01-16

Reject